# An ensemble-based approach for pumping optimization in an island aquifer considering parameter, observation and climate uncertainty

Cécile Coulon[1,2,3,4], Jeremy T. White[5], Alexandre Pryet[1,6], Laura Gatel[1,2], Jean-Michel Lemieux[1,2,3]

[1]Département de géologie et de génie géologique, Université Laval, Québec (Québec), G1V 0A6, Canada
[2]Centre québécois de recherche sur l'eau, Québec (Québec), G1V 0A6, Canada
[3]Centre d'études nordiques, Université Laval, Québec (Québec), G1V 0A6, Canada
[4]Current address: INTERA SAS, Limonest, France
[5]INTERA Geosciences Pty Ltd, Perth, Australia
[6]EPOC (UMR 5805), CNRS, Univ. Bordeaux & Bordeaux INP, France

*Correspondence to:* Cécile Coulon (ccoulon@intera.com)

**Abstract.** In coastal zones, a major objective of groundwater management is often to determine sustainable pumping rates which avoid well salinization. Understanding how model and climate uncertainties affect optimal management solutions is essential to provide groundwater managers information about salinization risk, and is facilitated by the use of optimization under uncertainty (OUU) methods. However, guidelines are missing for the widespread implementation of OUU in real-world coastal aquifers, and for the incorporation of climate uncertainty into OUU approaches. An ensemble-based OUU approach was developed, considering parameter, observation and climate uncertainty, and was implemented in a real-world island aquifer in the Magdalen Islands (Quebec, Canada). A sharp-interface seawater intrusion model was developed using MODFLOW-SWI2 and a prior parameter ensemble was generated, containing multiple equally-plausible realizations. Ensemble-based history matching was conducted using an iterative ensemble smoother which yielded a posterior parameter ensemble conveying both parameter and observation uncertainty. Sea level and recharge ensembles were generated for the year 2050, which were then used to generate a predictive parameter ensemble conveying parameter, observation and climate uncertainty. Multi-objective OUU was then conducted, aiming to both maximize pumping rates and minimize the probability of well salinization. As a result, the optimal trade-off between pumping and probability of salinization was quantified, considering parameter, historical observation and future climate uncertainty simultaneously. The multi-objective, ensemble-based OUU led to optimal pumping rates that were very different from a previous deterministic OUU, and close to the current and projected water demand for risk-averse stances. Incorporating climate uncertainty in the OUU was also critical since it reduced the maximum allowable pumping rates for users with a risk-averse stance. The workflow used tools adapted to very high-dimensional, nonlinear models and optimization problems, to facilitate its implementation in a wide range of real-world settings.

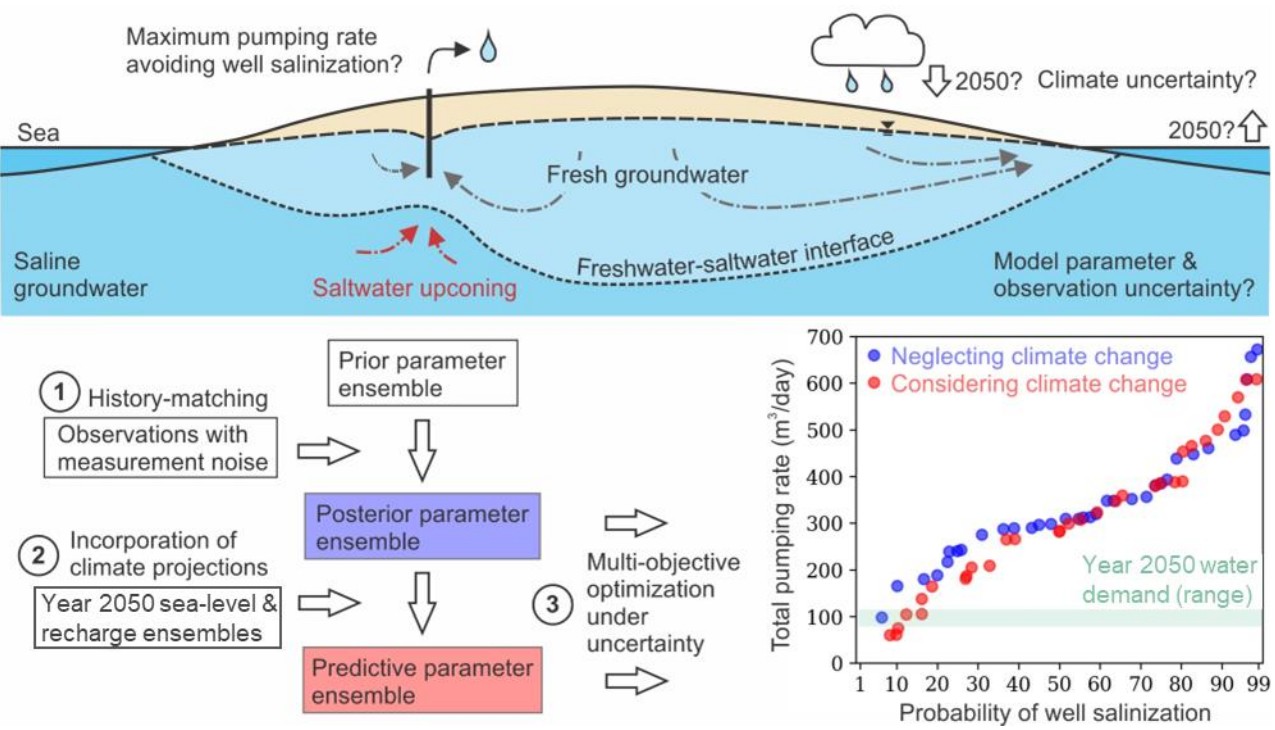

## 1 Introduction

Seawater intrusion is a major challenge for groundwater management in coastal zones, which are under pressure due to population growth, sea-level rise and changes in climate (Michael et al., 2017; Jiao and Post, 2019). Numerical models are often relied on to support groundwater management, using either advective-dispersive solute transport codes, which simulate mixing between freshwater and saltwater and are computationally demanding, or sharp-interface codes, which consider freshwater and saltwater to be immiscible but have significantly shorter simulation times. These numerical models can provide insight into freshwater availability under current and projected conditions, and are frequently combined with optimization algorithms to determine sustainable pumping rates which avoid well salinization (Ketabchi and Ataie-Ashtiani, 2015). To provide information to groundwater managers about salinization risk, optimizations in coastal zones should recognize the uncertainty in model predictions (Werner et al., 2013). However, because of the high computational costs associated with both advective-dispersive solute transport models and stochastic uncertainty quantification, optimization under uncertainty (OUU) approaches have mostly been implemented in synthetic or simplified real-world cases (Rajabi and Ketabchi, 2017; Mostafaei-Avandari and Ketabchi, 2020). Guidance is missing for the implementation of OUU in real-world coastal settings (Ketabchi and Ataie-Ashtiani, 2015).

Real-world coastal OUU applications have generally used advective-dispersive solute transport models in combination with surrogate models, evolutionary algorithms, and stochastic uncertainty quantification accounting for the uncertainty of a few model parameters (e.g., between 2 and 11 parameters in Sreekanth and Datta, 2014; Rajabi and Ketabchi, 2017; Lal and Datta, 2019; Mostafaei-Avandari and Ketabchi, 2020; Han et al., 2021). These applications were also not systematically preceded by parameter estimation (i.e., automated calibration). However, accounting for model parameter uncertainty accurately and robustly can require very high numbers of parameters (White, 2018). Furthermore, conducting data assimilation prior to OUU is essential, since it allows to update parameter uncertainty estimates with observation uncertainty. In Coulon et al. (2022), the real-world OUU approach used a sharp-interface model, sequential linear programming (SLP), and first-order-second-moment (FOSM) uncertainty analysis accounting for both parameter and observation uncertainty (including 60 parameters and 162 observations). The approach was preceded by parameter estimation (Coulon et al., 2021), conducted using the widely used Gauss-Levenberg-Marquardt (GLM) algorithm. However, the GLM and SLP algorithms can be computationally costly to apply in very highly parameterized and nonlinear models, which can limit the implementation of this workflow in other coastal settings. Furthermore, FOSM-based uncertainty analysis relies on strong assumptions of model linearity and Gaussian distributions of uncertainty, and therefore can only provide approximations of model predictive uncertainty. Stochastic, ensemble-based approaches are needed to provide more reliable uncertainty estimates. Developing an ensemble-based approach for coastal OUU adapted to highly-parameterized, nonlinear models would facilitate the implementation of OUU methods in real-world coastal settings.

Coupling climate uncertainty with model parameter uncertainty and understanding the consequences for groundwater management was identified as a prospective topic of seawater intrusion research (Werner et al., 2013). However, current OUU approaches have only considered parameter uncertainty (e.g. in hydraulic conductivity, recharge, porosity, longitudinal dispersivity – Mostafaei-Avandari and Ketabchi, 2020), while climate change projections were incorporated into optimizations as discrete scenarios (e.g., optimization under a projected sea-level rise scenario or under four projected recharge scenarios in Roy and Datta (2018) and Zhao et al. (2021), respectively). The uncertainty associated with climate projections has therefore not been considered, although in the field of hydrology, climate uncertainty is often evaluated using ensembles of climate projections (Mustafa et al., 2019; Al Atawneh et al., 2021). Generating ensembles of climate projections and combining them with model parameter ensembles, within an OUU framework, would enable evaluating the coupled impacts of climate and parameter uncertainty on groundwater management solutions, and lead to more robust decision-making.

The objective of this study was to provide a framework for stochastic, ensemble-based pumping optimization under uncertainty in an island aquifer, considering parameter, historical observation and future climate uncertainty, and using methods adapted to very high-dimensional, nonlinear models. An ensemble-based framework was implemented using a sharp-interface modeling approach. A sharp-interface model was first built using MODFLOW-SWI2 (Bakker et al., 2013), after which ensemble-based history matching was conducted using the iterative ensemble smoother PESTPP-IES (White, 2018). Sea level and recharge climate ensembles were generated for the year 2050 and incorporated into the model parameter ensemble. Multi-objective optimization under uncertainty was then conducted via PESTPP-MOU (White et al., 2022), using the ensemble to account for uncertainty in the simulated response to groundwater extraction within the optimization process. Results were compared both to the current water demand and to water demand projections for the year 2050. This workflow was applied to a real-world island aquifer in the Magdalen Islands (Quebec, Canada) and was entirely scripted in Python, to enhance the transparency and reproducibility of the analyses (e.g., White et al., 2020a; Fienen and Bakker, 2016).

## 2 Study area

Grande Entrée Island forms part of the Magdalen Islands archipelago, which is located in the middle of the Gulf of Saint-Lawrence (Fig. 1). Fresh groundwater is the only source of drinking water for the sparsely populated island communities, and is contained in a lens overlying saline groundwater. The pumping wells of the water supply system are at risk of salinization, since freshwater abstraction leads to the upward migration of saline groundwater towards the wells, a process called saltwater upconing. Deterministic parameter estimation and SLP, FOSM-based OUU approaches were previously implemented on Grande Entrée Island to determine maximum pumping rates which avoid well salinization, considering parameter and observation uncertainty and current climate conditions (Coulon et al., 2021; Coulon et al., 2022). This study details the implementation of the ensemble-based approach for history matching and multi-objective (MO) OUU, and the incorporation of climate uncertainty (specifically, sea level and recharge uncertainty) into the pumping optimization. Optimization results of the MO-ensemble approach are compared to those of the SLP-FOSM approach.

A detailed description of the study area is available in Coulon et al. (2021). The geology is comprised of a highly-permeable and heterogeneous Permian sandstone (Fig. 1), along with Quaternary sand dunes and glacial sediments (mostly fine sand). A spatially distributed recharge representative of the period 1989-2019 was generated for the Magdalen Islands archipelago by Lemieux et al. (2022), using a SWB2 groundwater recharge model (Soil-Water-Balance-2, Westenbroek et al., 2018). A spatial average of 524 mm/year was determined for the whole archipelago, and 559 mm/year was applied for Grande-Entrée Island specifically. Using quadratic regression and extrapolation of the Magdalen Islands' tide gage data, Barnett et al. (2017) projected a median relative sea-level rise of 0.19 m for the archipelago between 2020 and 2050, assuming a normal probability distribution for the 2050 projected sea level and a standard deviation of 0.11 m. Nine municipal pumping wells are located in the Permian sandstone formation, providing freshwater to approximately 2,800 inhabitants and to the commercial, industrial and institutional sectors since 2013. Between 2014 and 2020, the mean water demand in the network was approximately 93 $m^3$/day. A median water demand projection of 94 $m^3$/day was estimated for the year 2050, with a range of possible values between 76 and 115 $m^3$/day, showing no significant evolution in water demand but uncertainty in the projections (Lemieux et al., 2022).

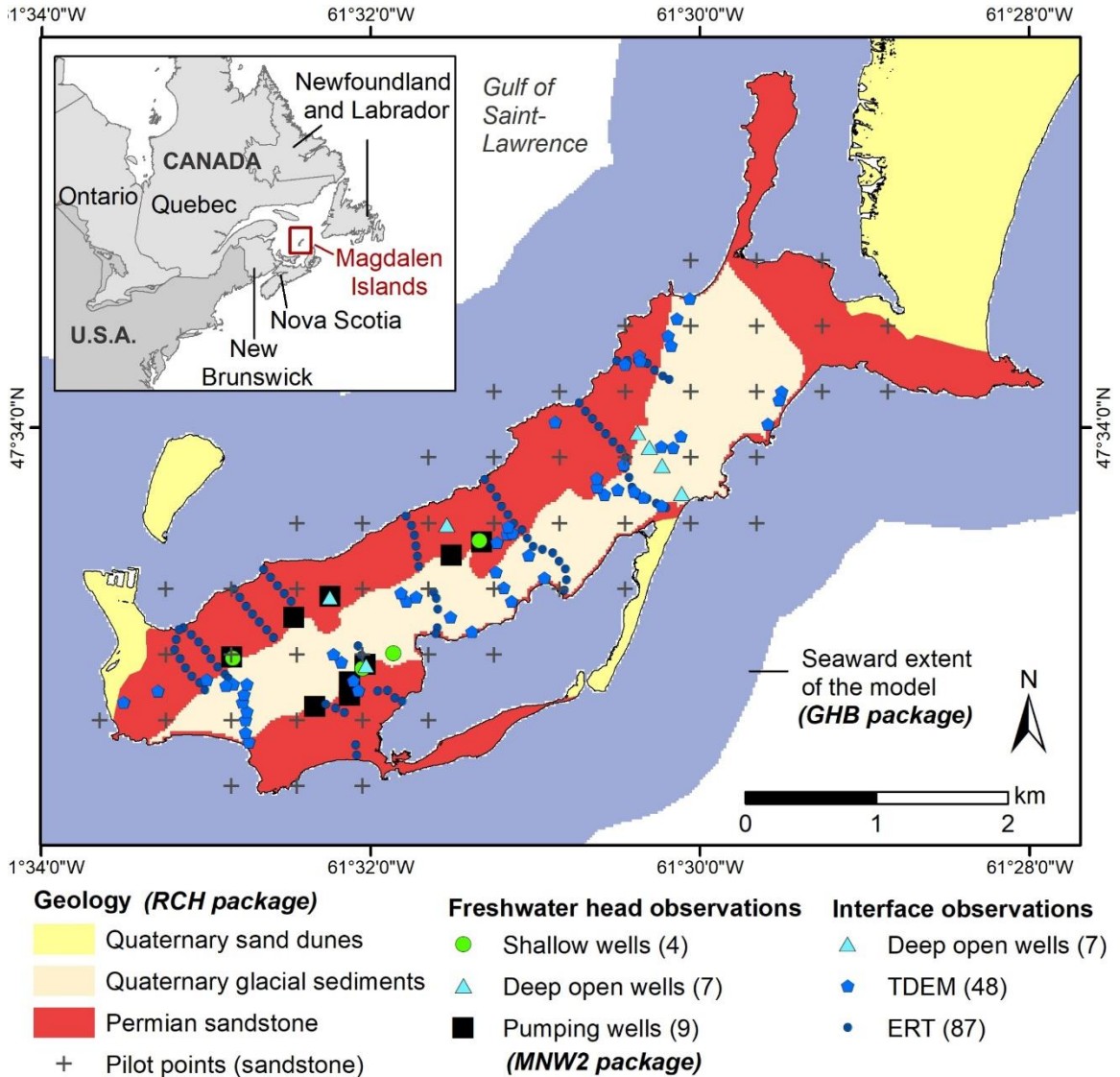

**Figure 1:** Map view of the Grande Entrée Island numerical model with seaward extent, geological formations and locations of pilot points, model observations (including electrical resistivity tomography and time-domain electromagnetic surveys) and pumping wells (domestic wells not shown). The boundary conditions implemented in MODFLOW are a uniform recharge rate on land cells (RCH package), general head boundary conditions for sea cells (GHB package), and groundwater pumping at municipal wells (MNW2 package). Modified from Coulon et al. (2022).

## 3 Methods

### 3.1 Numerical model

A sharp-interface, 2D-horizontal model with a 20 m x 20 m grid was developed using the SWI2 package (Bakker et al., 2013) for MODFLOW-2005 (Harbaugh, 2005), which simulates vertically-integrated variable-density groundwater flow but does not account for hydrodynamic dispersion. Within the single model layer, the groundwater was divided into a freshwater zone and a saltwater zone, separated by an interface representing the 50% seawater salinity contour. The numerical model is described in detail by Coulon et al. (2021). A general head boundary condition was implemented in the offshore cells, to convert the sea level into equivalent freshwater heads at the seabed (Bakker et al., 2013), and a uniform recharge rate was implemented on all land cells (Fig. 1). Municipal groundwater pumping was simulated using the MNW2 package (Konikow et al., 2009), to assimilate water levels (Coulon et al., 2021) and to calculate optimization constraints (Coulon et al., 2022). While domestic pumping was also simulated (Coulon et al., 2021), in the rest of the paper the term "pumping wells" will refer to the municipal pumping wells only.

The hydraulic conductivity field was parameterized using a combination of pilot points and zones of piecewise constancy (Doherty, 2003). As part of the pilot-point parameterization, ordinary kriging was used to interpolate grid cell values from pilot point locations to the model cells. A homogeneous transverse dispersivity was implemented as a correction factor for the sharp interface (Coulon et al., 2021), while two additional parameters, longitudinal dispersivity and the initial transition zone width, were used in the calculation of the optimization constraint (Coulon et al., 2022). Apart from recharge, whose prior value was updated with the Lemieux et al. (2022) estimates (Section 2), all model parameters were assigned prior values and ranges equal to those in Coulon et al. (2021) and Coulon et al. (2022), based on field measurements and existing literature, and assuming normal or lognormal probability distributions (Table 1). This allowed for the comparison between the previous SLP-FOSM approach and the current MO-ensemble approach.

**Table 1:** Prior parameter probability distributions of the uncertain model parameters, assumed to be normal or lognormal, described by the mean and the 95% confidence interval (i.e. the mean $\pm$ 2 times the standard deviation).

| | Mean | 95% confidence interval |
|---|---|---|
| $K_{\text{sand dunes}}$ (m/s) | $5 \times 10^{-3}$ | $5 \times 10^{-5} - 5 \times 10^{-1}$ |
| $K_{\text{sandstones}}$ (m/s) (offshore) | $4 \times 10^{-5}$ | $3 \times 10^{-6} - 6 \times 10^{-4}$ |
| $K_{\text{sandstones}}$ (m/s) (52 pilot points) | | |
| $K_{\text{glacial sediments}}$ (m/s) | $1 \times 10^{-5}$ | $1 \times 10^{-7} - 1 \times 10^{-3}$ |
| $K_{\text{seabed}}$ (m/s) | $2 \times 10^{-5}$ | $2 \times 10^{-7} - 2 \times 10^{-3}$ |
| Transverse vertical dispersivity $\alpha_{\text{T}}$ (m) | $1 \times 10^{-1}$ | $1 \times 10^{-3} - 10$ |
| Recharge (mm/yr) | 560 | $360 - 760$ |
| Initial transition zone width $M$ (m) | 8 | $5 - 11$ |
| Longitudinal dispersivity $\alpha_{\text{L}}$ (m) | 3 | $1 - 5$ |

### 3.2 Ensemble-based history matching

Instead of adopting a deterministic approach, which seeks the minimum error variance parameter set resulting in the best fit to observations, an ensemble-based approach was selected in which multiple, equally-plausible realizations of parameter sets were generated (i.e., a parameter ensemble) and carried forward into the analyses (Anderson et al., 2015). The following definitions will be used throughout the paper. A realization is a parameter set; collectively, many realizations form a parameter ensemble. The prior parameter ensemble is the set of realizations that is generated prior to history matching, using only expert knowledge to define the statistical distributions of parameters. The posterior parameter ensemble is the set of realizations obtained after history matching (i.e., the prior parameter ensemble updated by observations). The predictive parameter ensemble is the set of realizations obtained after incorporating climate ensembles into the posterior parameter ensemble, all other aquifer parameters remaining constant (i.e., the posterior parameter ensemble updated by climate projections). The size of an ensemble corresponds to the number of realizations comprising the ensemble. The term "ensemble" is used in line with the PEST++ terminology (White et al., 2020b), and has the same meaning as the term "stack" commonly used in OUU literature (Bayer et al., 2008). Figure 2 summarizes the ensemble-based framework that was developed.

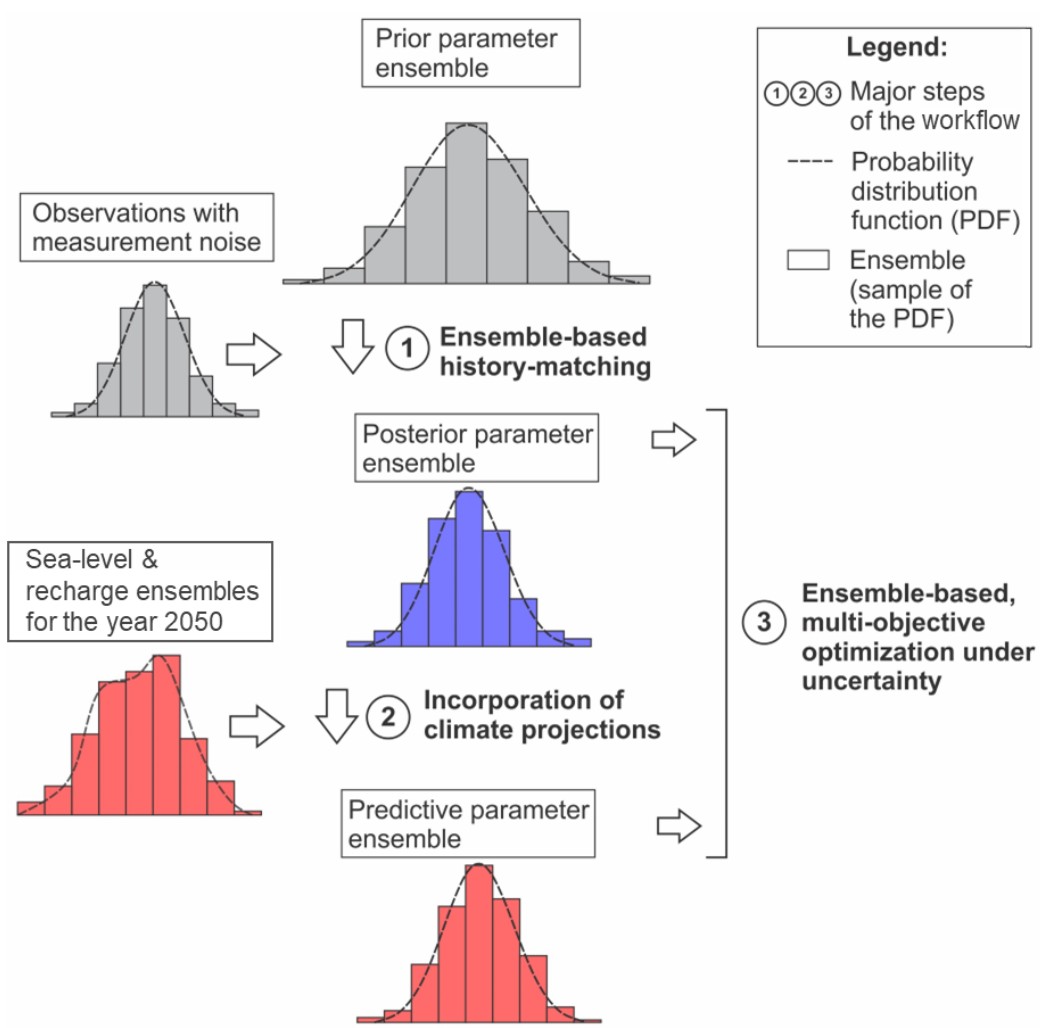

**Figure 2:** Summary of the ensemble-based workflow. An ensemble represents a sample of the probability distribution function (PDF). After generating a prior parameter ensemble by drawing from prior parameter PDFs, 1) ensemble-based history matching yielded a posterior parameter ensemble; 2) a predictive parameter ensemble was obtained by updating the posterior parameter ensemble with year 2050 sea level and recharge projections; and 3) optimization under uncertainty was conducted using both the posterior and the predictive parameter ensembles, to obtain maximum allowable pumping rates considering parameter and observation uncertainty and either neglecting or accounting for climate projections, respectively.

Fifty-eight model parameters were considered to be adjustable, including 56 hydraulic conductivity values, a uniform recharge rate and a homogeneous transverse dispersivity (Section 3.1). $N_{prior}$ random realizations were drawn from the prior probability distribution functions (PDFs) of these parameters (Table 1), assuming they could be described by multi-Gaussian distributions and using a prior parameter covariance matrix. It was assumed that all parameters were statistically independent, except for pilot point parameters which were spatially correlated. To describe the spatial correlation between the hydraulic conductivities at pilot point locations, an exponential variogram with a range equal to 3 times the pilot point spacing (i.e., 500 m) was used. We note that parameters varying over several orders of magnitude were all log-transformed. The prior parameter ensemble represented a sample of the prior parameter PDF (Fig. 2). Several constant model parameters were included in this ensemble and remained fixed during history matching, including the sea level, longitudinal dispersivity $\alpha_L$ and the initial width of the transition zone $M$ (Table 2). The dispersivity $\alpha_L$ and transition zone width $M$ were fixed during history matching but adjustable during optimization, to be consistent with the previous SLP-FOSM approach and to enable a comparison between both approaches.

Ensemble-based history matching was then conducted using PESTPP-IES (White, 2018), which implements an iterative ensemble smoother form of the GLM algorithm (Chen and Oliver, 2013). PESTPP-IES was selected because it enables history matching of

highly-parameterized models with a significantly lower computational effort than sampling methodologies such as Markov Chain Monte Carlo. History matching was implemented in a Bayesian framework, using the assumptions of multivariate Gaussian prior and posterior distributions. Over successive iterations, PESTPP-IES conditioned the prior parameter ensemble with the information contained in 20 freshwater head observations (extracted from shallow wells, deep open wells and pumping wells) and 142 freshwater-seawater interface elevation observations (derived from deep open wells, TDEM and ERT geophysical surveys, Fig. 1) by minimizing a model-to-measurement fit objective function. Observations were paired with random realizations of measurement noise, and the least-squares objective function was calculated as the sum of squared weighted differences between simulated and observed data (PEST++ Development Team, 2022). PESTPP-IES was limited to two iterations as an optimal trade-off between parameter posterior variance and fit to observations (Fienen et al., 2022). Details on the observation dataset and the measurement noise are provided by Coulon et al. (2021). History matching was conducted under steady-state conditions, using long transient (500-year) simulations with constant boundary conditions (i.e., sea level, recharge and pumping rates) representative of the average conditions during the 2014–2019 calibration period. The simulations were run until heads and interface elevations close to pumping wells were stable, which was achieved within 500 years. The model simulation times were approximately 8 minutes on a laptop computer (1.9 GHz Intel Core i7®). History matching yielded a posterior parameter ensemble of size $N_{post}$, representing a sample of the posterior parameter PDF (Fig. 2). $N_{post}$ is usually less than or equal to $N_{prior}$, since the parameter realizations resulting in model run failures and/or excessive simulation times are removed from the parameter ensemble during history matching. Running model simulations through the posterior parameter ensemble yielded posterior prediction ensembles, representing samples of the posterior prediction PDFs (Fig. 2) and conveying parameter and observation uncertainty for current conditions.

**Table 2:** Prior, posterior and predictive parameter ensembles, described by the ensemble size, the mean and the 5–95 percentile range. Statistics of the prior parameter ensemble are not identical to those of the prior parameter PDFs, because ensembles represent a sample of probability distributions. During history matching, the sea level, $M$ and $\alpha_L$ were fixed to constant values. The recharge and sea-level ensembles were updated in the predictive parameter ensemble. For the OUU, $M$ and $\alpha_L$ were considered uncertain in both the posterior and the predictive parameter ensembles. The percentile range of the pilot point hydraulic conductivities is not reported since it is different for each pilot point.

| Parameter | Prior parameter ensemble (size $N_{prior}$) | | Posterior parameter ensemble (size $N_{post}$) | | Predictive parameter ensemble (size $N_{post}$) | |
|---|---|---|---|---|---|---|
| | Mean | $5^{th} - 95^{th}$ percentiles | Mean | $5^{th} - 95^{th}$ percentiles | Mean | $5^{th}$-$95^{th}$ percentiles |
| $K_{sand\ dunes}$ (m/s) | $4 \times 10^{-2}$ | $1 \times 10^{-4} - 3 \times 10^{-1}$ | $7 \times 10^{-3}$ | $9 \times 10^{-5} - 4 \times 10^{-2}$ | Identical to the posterior ensemble | |
| $K_{sandstones}$ (m/s) (offshore) | $7 \times 10^{-5}$ | $4 \times 10^{-6} - 2 \times 10^{-4}$ | $1 \times 10^{-4}$ | $2 \times 10^{-5} - 5 \times 10^{-4}$ | | |
| $K_{sandstones}$ (m/s) (52 pilot points) | $8 \times 10^{-5}$ | – | $2 \times 10^{-4}$ | – | | |
| $K_{glacial\ sediments}$ (m/s) | $8 \times 10^{-5}$ | $1 \times 10^{-7} - 4 \times 10^{-4}$ | $5 \times 10^{-5}$ | $5 \times 10^{-7} - 2 \times 10^{-4}$ | | |
| $K_{seabed}$ (m/s) | $2 \times 10^{-4}$ | $4 \times 10^{-7} - 1 \times 10^{-3}$ | $1 \times 10^{-5}$ | $4 \times 10^{-6} - 3 \times 10^{-5}$ | | |
| Transverse vertical dispersivity $\alpha_T$ (m) | $9 \times 10^{-1}$ | $2 \times 10^{-3} - 5$ | $2 \times 10^{-2}$ | $2 \times 10^{-3} - 5 \times 10^{-2}$ | | |
| Recharge (mm/yr) | 547 | 371 – 696 | 572 | 444 – 695 | 574 | 344 – 826 |
| Sea level (masl) | 0.014 | – (fixed) | 0.014 | – (fixed) | 0.21 | 0.052 – 0.39 |
| Initial transition zone width $M$ (m) | 8 | – (fixed) | Identical to the prior ensemble for history matching, to the predictive ensemble for OUU | | 8 | 6 – 10 |
| Longitudinal dispersivity $\alpha_L$ (m) | 3 | – (fixed) | | | 3 | 1 – 5 |

### 3.3 Incorporating climate projections

The sea level and recharge ensembles contained in the posterior parameter ensemble were representative of the 2014–2019 calibration period (Table 2). To make model predictions for the year 2050, while accounting for climate projections, the ensembles were replaced with year 2050 sea level and recharge ensembles (Fig. 2). The climate change predictive simulations were then conducted under steady-state conditions, with boundary conditions representative of the 2050 sea-level and recharge conditions. All predictive simulations were conducted under steady-state conditions because the storage parameters were unconstrained by the history matching, and therefore remained highly uncertain. Running model simulations through the predictive parameter ensemble yielded prediction ensembles for the year 2050 conveying parameter, observation and climate uncertainty.

### 3.3.1 Sea-level ensemble

The reference elevation used for the numerical model was the local mean sea level of the Magdalen Islands (Lemieux et al., 2022), therefore the term "meters above sea level" (or "masl") is used in reference to this elevation. Using the relative sea-level rise projections for the study area (Section 2) and the current sea level (0.014 masl), a 0.204 masl mean sea level was expected for 2050, with a normal distribution and a standard deviation of 0.11 m. $N_{post}$ realizations were drawn from this probability distribution, from which values above 0 masl were truncated, to generate a sea-level ensemble for the year 2050. The current sea level used in the posterior parameter ensemble was replaced by the 2050 sea-level ensemble in the predictive parameter ensemble (Table 2).

### 3.3.2 Recharge ensemble

Seventy-two projections of daily precipitation, minimum and maximum air temperature were provided by the OURANOS consortium for the Magdalen Islands, for the period 2021–2050 (Charron, 2016). These resulted from three emission scenarios (the Representative Concentration Pathways RCP2.6, RCP4.5 and RCP8.5) being run through an ensemble of 24 Global Climate Models (GCMs) of the Coupled Model Intercomparison Project – Phase 5 (CMIP5). These 72 climate projections were run through the SWB2 groundwater recharge model developed by Lemieux et al. (2022) for the Magdalen Islands (Section 2), assuming a constant land use, which generated 72 daily recharge projections for the period 2021–2050 (Fig. 3a).

A recharge ensemble for the year 2050 was then extracted ($R_{2050,\ SWB2}$), containing 72 plausible recharge projections for 2050 generated by a physically-based groundwater recharge model (Section 2), and conveying climate uncertainty (specifically, the uncertainty in future emission scenarios and the inter-model uncertainty of the GCMs) (Fig. 3b). History matching yielded a current recharge ensemble ($R_{current,\ MODFLOW}$), containing $N_{post}$ plausible recharge inputs for the groundwater model (both informed by the observation dataset and incorporating possible correlations with the hydraulic conductivity parameters), and conveying parameter and observation uncertainty. The information contained in both ensembles needed to be merged, to obtain a 2050 recharge ensemble ($R_{2050,\ MODFLOW}$) containing $N_{post}$ plausible recharge inputs for the groundwater model, preserving the information acquired through history matching (i.e., fit to observations, parameter correlations) but also accounting for future climate projections, and therefore conveying parameter, observation and climate uncertainty.

A $\Delta R$ ensemble was first generated, containing 72 possible recharge variations between current conditions and 2050, i.e., 72 possible perturbations to current recharge conditions (Fig. 3c), by subtracting the average current recharge value estimated by SWB2, $R_{current,\ SWB2}$, (Section 2) from each of the projections in the $R_{2050,\ SWB2}$ ensemble:

$$\Delta R = R_{2050,\ SWB2} - R_{current,\ SWB2} \tag{1}$$

The $\Delta R$ ensemble was resampled to $N_{post}$ realizations assuming a normal distribution (Fig. 3d). The $\Delta R$ and $R_{current,}$

MODFLOW realizations were then randomly paired together to generate the $R_{2050,\ MODFLOW}$ ensemble :

$$R_{2050,\ \text{MODFLOW}} = R_{current,\ \text{MODFLOW}} + \Delta R \tag{2}$$

The $R_{current,\ MODFLOW}$ ensemble contained in the posterior parameter ensemble was replaced by $R_{2050,\ MODFLOW}$ in the predictive parameter ensemble (Table 2). The relative changes in projected recharge ($\Delta R$) were therefore used rather than absolute recharge projections ($R_{2050,\ SWB2}$). One of the major assumptions of this approach was that the recharge perturbations $\Delta R$ are uncorrelated

with the current recharge value. The sea level and recharge projections for the year 2050 were also assumed to be independent.

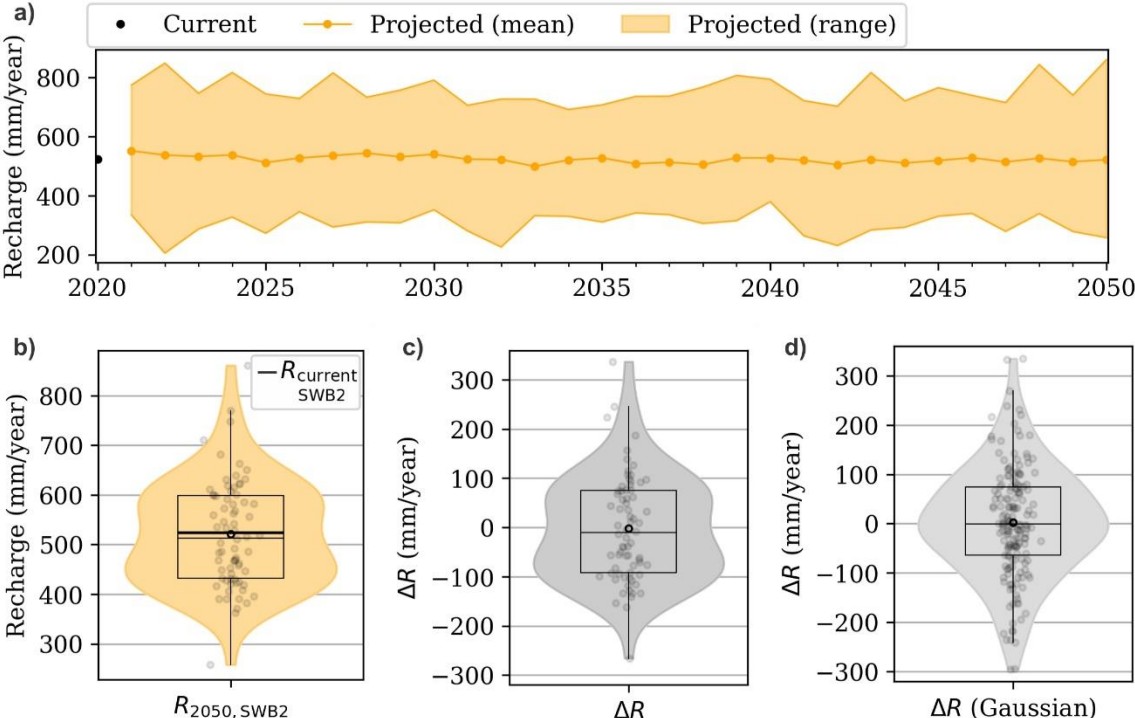

**Figure 3:** Processing of the climate ensembles generated by the SWB2 groundwater recharge model: (a) SWB2 recharge projections for 2021–2050; (b) $R_{2050,\ SWB2}$ ensemble extracted in 2050 (72 realizations); (c) $\Delta R$ ensemble obtained by subtracting $R_{current,\ SWB2}$ from $R_{2050,\ SWB2}$ (72 realizations); (d) $\Delta R$ ensemble resampled to $N_{post}$ realizations (i.e., the size of the posterior parameter ensemble) assuming a normal

distribution. In panels (b), (c) and (d) the data points, mean values (bold black circles) and box plots are superimposed onto the violin plots.

### 3.4 Optimization using ensemble-based uncertainty

The primary objective of the optimization was to maximize pumping rates in the well field while avoiding well salinization due to upconing. Salinization occurred if the 1% seawater salinity contour simulated under the well ($\zeta_{1\%}$), referred to as the optimization constraint, reached the well bottom elevation ($z_{botm}$). $\zeta_{1\%}$ was obtained by postprocessing the 50% seawater salinity contour

simulated by the groundwater model ($\zeta_{50\%}$), which introduced two new uncertain parameters, namely longitudinal dispersivity ($\alpha_L$) and the initial width of the transition zone ($M$) (see Coulon et al., (2022) for more details). An additional objective was introduced to the optimization problem, i.e., maximization of reliability, effectively converting the single-objective optimization into a reliability-based, two-objective optimization (Deb et al., 2007). In an ensemble-based framework, running a single pumping scenario through a parameter ensemble generates a constraint ensemble, in which the constraints can be satisfied (i.e., well

salinization is avoided) for a fraction of the realizations, while constraints are violated (i.e., well salinization has occurred) in the others. The reliability $Re$ is the probability of the constraints being satisfied in all of the realizations in the ensemble (Bayer et al., 2008), i.e., the probability of avoiding well salinization simultaneously for all wells. $100 - Re$ represents the probability of well

salinization. Multiple optimal pumping scenarios can be determined for different reliability values, depending on the degree of tolerance towards risk (Fig. 4). The OUU was mathematically formulated as a constrained two-objective optimization:

$$\text{Maximize } Q_{\text{total}} = \sum_{i}^{n} Q_i, \qquad (i = 1, \dots, n) \qquad (3)$$

$$\text{Maximize } Re = P(\zeta_{1\% \, i} \leq z_{\text{botm} \, i}) \qquad (i = 1, \dots, n) \qquad (4)$$

$$\text{Subject to } \zeta_{1\% \, i} \leq z_{\text{botm} \, i} \qquad (i = 1, \dots, n) \qquad (5)$$

$$\text{and } Q_{\text{min} \, i} \leq Q_i \leq Q_{\text{max} \, i} \qquad (i = 1, \dots, n) \qquad (6)$$

where $Q_{\text{total}}$ is the total pumping rate in the well field (m³/day), $n$ is the number of pumping wells (nine in total), $Q_i$ is the pumping rate at each well $i$ (m³/day), i.e., the decision variables of the optimization, $Q_{\text{min} \, i}$ and $Q_{\text{max},i}$ are the minimum and maximum pumping rates at each well $i$, respectively (m³/day), $Re$ is the reliability, $\zeta_{1\% \, i}$ is the elevation of the 1% seawater salinity contour under each well $i$ (masl), i.e, the optimization constraints, and $z_{\text{botm} \, i}$ is the bottom elevation of well $i$ (masl). Wide $Q_{\text{min} \, i}$ and $Q_{\text{max},i}$ values were set, which effectively removed the constraint presented in Eq. (6).

Parameters $M$ and $\alpha_L$, which had remained fixed during history matching (Section 3.2), were considered to be uncertain in the optimization (Coulon et al., 2022). $N_{\text{post}}$ realizations were drawn from the prior PDFs of $M$ and $\alpha_L$ (Table 1) and were spliced into the posterior and predictive parameter ensembles (Table 2). Long transient (500-year) initial simulations with no pumping were run through both parameter ensembles, to allow the freshwater lens to reach a steady state condition under the climate forcings and hydraulic properties prescribed in each realization. The pumping optimization under uncertainty was then conducted under steady-state conditions, using long transient (200-year) simulations to allow the freshwater lens to reach a new steady state under the tested pumping rates. The occurrence of well salinization was examined at the end of the 200-year simulation period. Model simulation times were approximately 5 minutes on a laptop computer (1.9 GHz Intel Core i7®).

The optimization problem was solved using the NSGA-II nondominated-sorting genetic algorithm (Deb et al., 2002) implemented in PESTPP-MOU (White et al., 2022), using ensemble-based constraint uncertainty. PESTPP-MOU was selected because it implements a wide range of evolutionary algorithms, which are more effective than traditional optimization methods when solving highly nonlinear optimization problems typical of coastal environments (Ketabchi and Ataie-Ashtiani, 2015), its use required very little modification to input files after having previously used the PEST++ software, and the reliability-based optimization could be implemented using PESTPP-MOU's "risk as an objective" option (White et al., 2022). The NSGA-II algorithm has been applied in many past coastal OUU studies (Mostafaei-Avandari and Ketabchi, 2020) , and uses a population-based approach to identify the Pareto front, which is the optimal trade-off surface between competing objectives (on which any further improvement to one of the objectives results in the reduction of another). The NSGA-II algorithm was implemented using a population size of 30 and 150 generations. At each generation, PESTPP-MOU generated new individuals (i.e., new combinations of decision variables) from the parent population using differential evolution (Storn and Price, 1997), ranked all individuals according to their fitness using the NSGA-II algorithm (Deb et al., 2002), and selected the most fit individuals to be the new parent population for the next generation (details in Fig. 5). The prediction ensemble was evaluated for all individuals of the initial population, re-evaluated every 10 generations (which each required a total of 30 x $N_{\text{post}}$ model simulations) and reused in the intermediate generations, as a trade-off between uncertainty quantification and computational constraints. In intermediate generations, each individual was mapped to the nearest individual at which constraint PDFs had been previously evaluated, in a minimum-Euclidean-distance sense, and the constraint PDFs of the latter were translated to the former. This approach assumes that individuals close to each other in decision variable space have similar constraint PDFs (White et al., 2022; PEST++ Development Team, 2022).

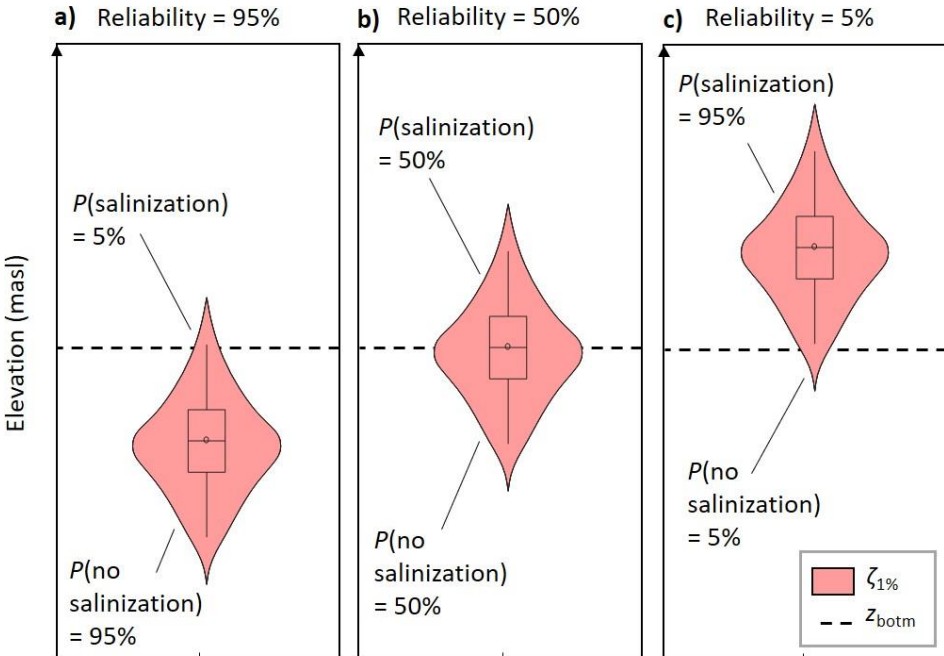

**Figure 4:** Schematic 1% seawater salinity contour ($\zeta_{1\%}$) constraint ensembles, represented by violin plots and box plots, resulting from optimizations with reliabilities of (a) 95%, (b) 50% and (c) 5%. Panels (a), (b) and (c) correspond to probabilities of well salinization of 5% (risk-averse stance), 50% (risk-neutral stance) and 95% (risk-tolerant stance), respectively. Modified from Coulon et al. (2022).

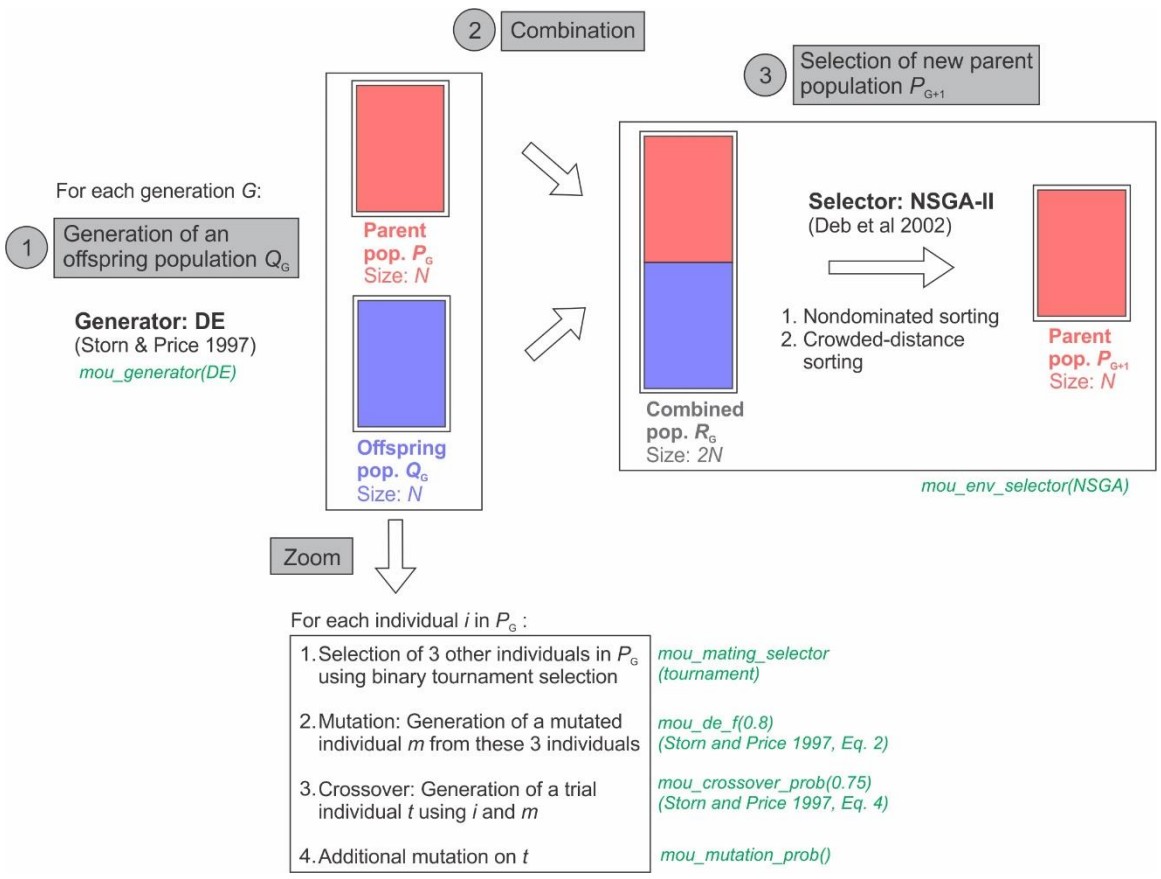

**Figure 5:** Summary of the main steps undertaken in each generation of the optimization, as a result of all PESTPP-MOU parameters (in green) being set to their default values (modified from Deb et al., 2002). The handling of chance constraints is not shown. More information on differential evolution (DE), the nondominated sorting genetic algorithm NSGA-II and PESTPP-MOU can be found in Storn and Price (1997), Deb et al. (2002) and PEST++ Development Team (2022), respectively.

## 4 Results

### 4.1 History matching

As the number of realizations increases, the ensembles become more representative of the PDFs that they sample, but computational times increase. The most important factor influencing the outcome of the OUU procedure is whether the statistical moments of the constraint ensembles have converged relative to the ensemble size, since the probabilities of constraint violation are directly used in the optimization algorithm (Eq 4, 5). Analyzing the convergence of posterior $\zeta_{50\%}$ ensemble mean and standard deviation values as a function of prior ensemble size led to the selection of a prior parameter ensemble containing 200 realizations. History matching required 977 model runs, which took 2 hours using 50 cores at 2.3 GHz. During history matching (Section 3.2), 27 realizations were abandoned, resulting in a posterior parameter ensemble containing 173 realizations. The final model-to-measurement fit objective function (Section 3.2) was in the range of 1000–3300, with a mean value of 1250. An acceptable model-to-measurement misfit was obtained both for freshwater heads (Fig. 6a) and freshwater-seawater interface elevations (Fig. 6b), and no prior-data conflicts were observed. These results were similar to those of the deterministic parameter estimation conducted by Coulon et al. (2021), which had yielded a final objective function of 1175 and similar patterns in the simulated-to-observed scatterplots (e.g., an outlier in the municipal well dataset and biased freshwater head observations in deep wells). Hydrogeologically reasonable values were obtained in the posterior parameter ensemble, with the sand dunes generally more permeable than the sandstones, which were in turn more permeable than the glacial sediments (Table 2). Figure 7 presents a random sample of the 173 hydraulic conductivity fields obtained through history matching, and the associated recharge values.

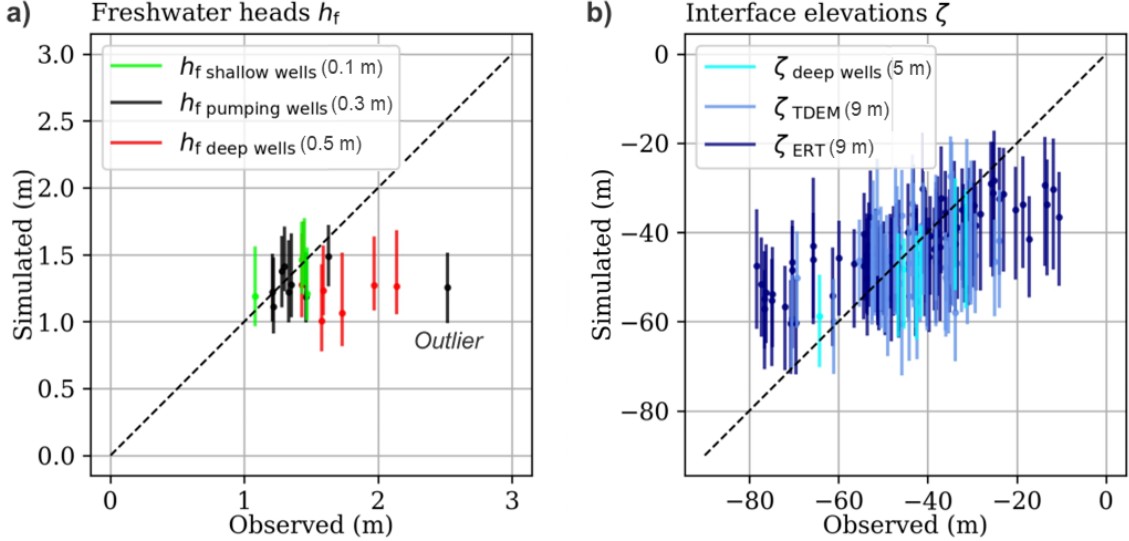

**Figure 6:** Scatter plots of observed vs. simulated data, for (a) freshwater heads and (b) freshwater-seawater interface elevations. Each observation corresponds to an ensemble of simulated values: the line extends from the minimum to the maximum value and the point represents the mean. The 1:1 diagonal line represents equal simulated and observed data. The average MAE value (mean average error) for each observation group is shown.

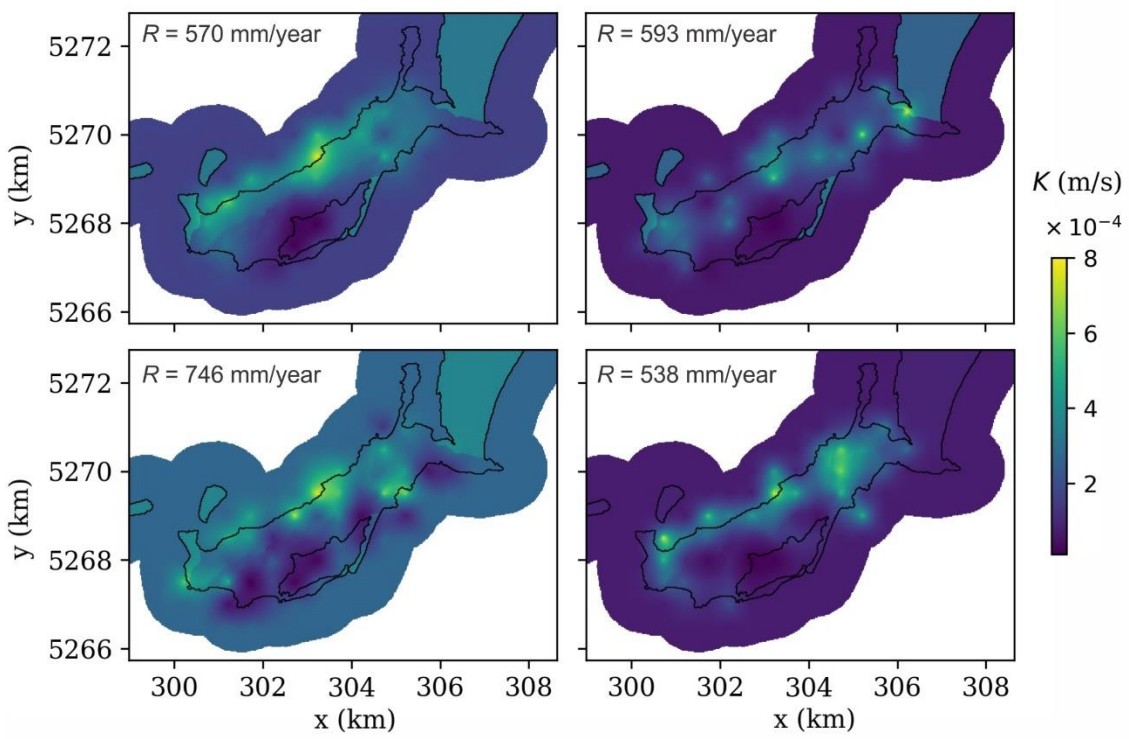

**Figure 7:** Example of four realizations extracted from the posterior parameter ensemble. Posterior hydraulic conductivity ($K$) fields and the associated recharge values ($R$) are shown.

### 4.2 Incorporating climate projections

All projections in the year 2050 sea-level ensemble were greater than the current mean sea level (Fig. 8a), with a mean projected sea-level rise of 0.2 m and an uncertainty (represented by the standard deviation) of 0.1 m. The $\Delta R$ ensemble, i.e., the ensemble containing possible recharge variations between current conditions and 2050 (Section 3.3.2), showed little-to-no evolution in the mean recharge (mean value close to zero, Fig. 3c), but had a climate-related uncertainty of 108 mm/year. As a consequence, the current and 2050 recharge ensembles had very similar median values (less than 1% variation, Fig. 8b), but the uncertainty of the

2050 recharge ensemble increased by 86% (from 78 to 145 mm/year). The $\zeta_{50\%}$ ensembles obtained using the posterior and predictive parameter ensembles, i.e., when neglecting or accounting for climate projections, respectively, were compared at steady-state conditions with zero pumping (i.e., the pumping optimization initial conditions, Section 3.4). Figure 9 shows the variability in the pumping optimization initial conditions obtained with both ensembles. Both ensembles had very similar median values (less than 0.5% variation on average, Fig. 9). However, the uncertainty of the posterior $\zeta_{50\%}$ ensembles increased significantly (on

average by 114%, from 4 to 8 m). When accounting for climate projections, some realizations in the predictive parameter ensemble led to $\zeta_{50\%}$ values reaching well bottom elevations, even with zero pumping (e.g. at wells no. 7 and 8, Fig. 9).

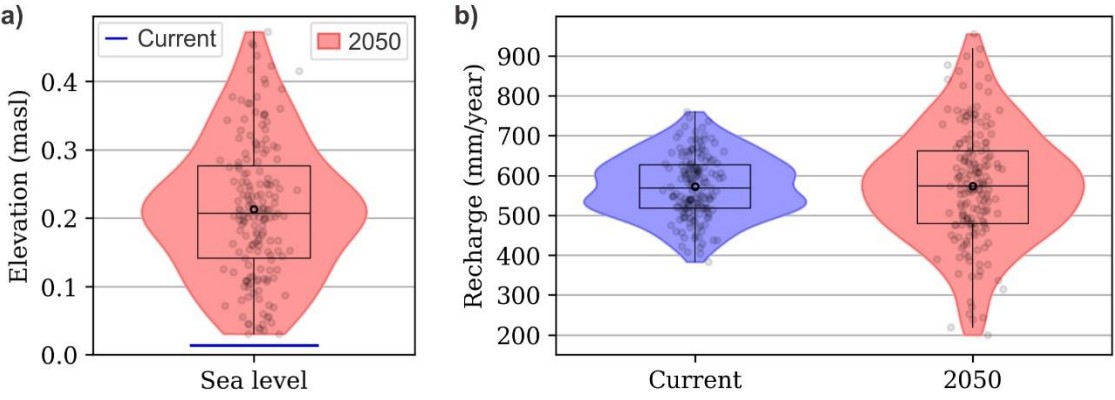

**Figure 8:** Current vs year 2050 projected (a) sea level and (b) recharge ensembles, as implemented in either the posterior (blue) or the predictive (red) parameter ensembles. Data points, mean values (bold black circle) and box plots are superimposed on the violin plots.

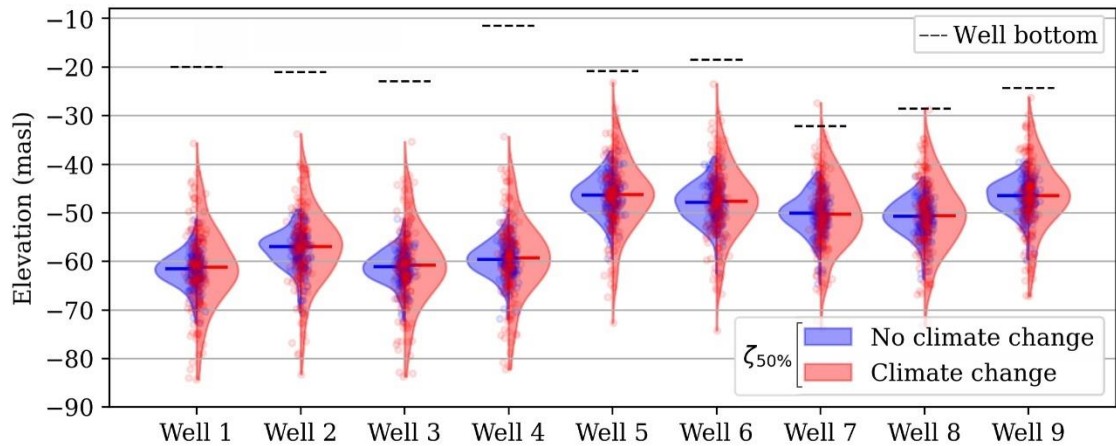

**Figure 9:** 50% seawater salinity contour ($\zeta_{50\%}$) ensembles under each pumping well, under steady-state conditions without pumping, when neglecting (blue) or accounting for (red) climate projections. Data points, median values (horizontal lines) and well bottom elevations are superimposed on the violin plots.

### 4.3 Optimization under uncertainty

The OUU procedure for the posterior and predictive parameter ensembles required approximately 89,000 and 86,000 model simulations, respectively, which took a total of 320 hours, using 130 cores at 2.3 GHz. 34 and 35 Pareto-optimal pumping scenarios were identified. The Pareto front, or optimal trade-off between pumping and probability of well salinization, was identified (Fig. 10), and the optimal allocation of pumping in the well field was determined for a range of probabilities of well salinization (Fig. 11). As the maximum pumping rate in the well field (Fig. 10) and at individual wells (Fig. 11) increased, the probability of well salinization increased as well.

The results of the MO-ensemble approach (when neglecting climate projections) were first compared to those of the SLP-FOSM approach described in Coulon et al. (2022). The maximum allowable pumping rates determined by the MO-ensemble approach were lower for all but highly risk-tolerant stances (Fig. 10), and the difference between both approaches was largest at highly risk-averse stances. For example, for a 6% probability of well salinization, the MO-ensemble approach found a maximum allowable pumping rate of 98 m³/day, versus 260 m³/day for the SLP-FOSM approach. For risk-averse stances, the pumping rates at individual wells were generally lower (e.g., wells 1 and 5, Fig. 11). The MO-ensemble approach determined higher probabilities of well salinization than the SLP-FOSM approach, for equal values of total pumping in the well field. For example, a 230 m³/day pumping rate corresponded to a 25% probability of salinization using the MO-ensemble approach, versus 2% using the SLP-FOSM

approach. In fact, the MO-ensemble approach did not identify any pumping scenarios with probabilities of salinization under 6% (Fig. 10). Most importantly, the most risk-averse pumping scenario found by the MO-ensemble approach (98 m³/day) was of the same order of magnitude as both the current and projected water demand, whereas the SLP-FOSM approach systematically found pumping scenarios far greater than the water demand.

The consequences of neglecting or accounting for climate projections within the MO-ensemble approach were then examined. When climate projections were considered, the maximum allowable pumping rates were lower for risk-averse stances (Fig. 10). Therefore, when considering climate projections, pumping rates needed to be lowered to conserve identical probabilities of well salinization; for example, selecting a 10% probability of salinization required reducing the total pumping in the well field from 165 to 60 m³/day (Fig. 10), and the pumping rate at well no. 8 from 19 to 4 m³/day (Fig. 11). On the other hand, neglecting climate projections resulted in underestimating probabilities of salinization, since a 98 m³/day total pumping rate in the well field represented a 6% probability of salinization when neglecting climate projections, but a 12% probability of salinization when considering climate projections (Fig. 10). The dispersion and non-unicity of the individual pumping rates identified by the ensemble-based method (Fig. 11) made a detailed comparison of individual pumping rates challenging. Overall, when considering climate projections, no pumping scenarios were found with probabilities of well salinization lower than 8%. Pumping rates under the most risk-averse scenarios (with probabilities of salinization between 8–10%) were less than both the current and projected water demand, while rates under less risk-averse scenarios (with probabilities of salinization between 10–15%) were of the same order of magnitude as the water demand.

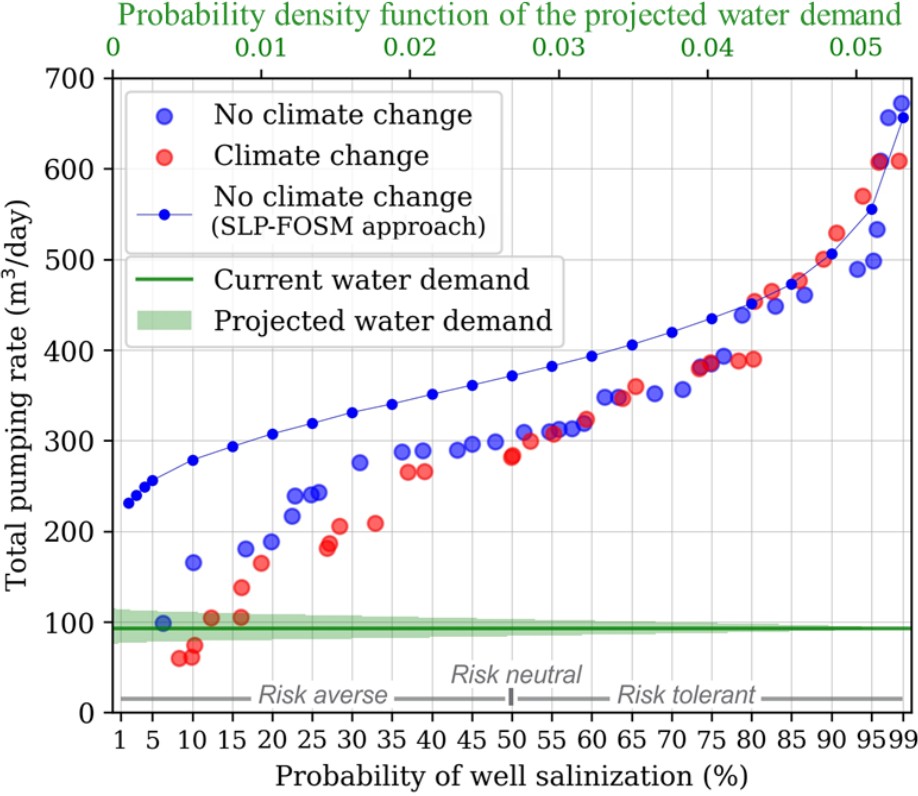

**Figure 10:** Optimal trade-off between the total pumping rate in the well field and the probability of well salinization, when neglecting (blue) or accounting for (red) climate projections (ensemble-based approach). The results obtained using the SLP-FOSM approach (neglecting climate change effects) are also shown. The current water demand and the probability density function associated with the projected (year 2050) water demand are superimposed. The y-axis was cut off at 700 m³/day.

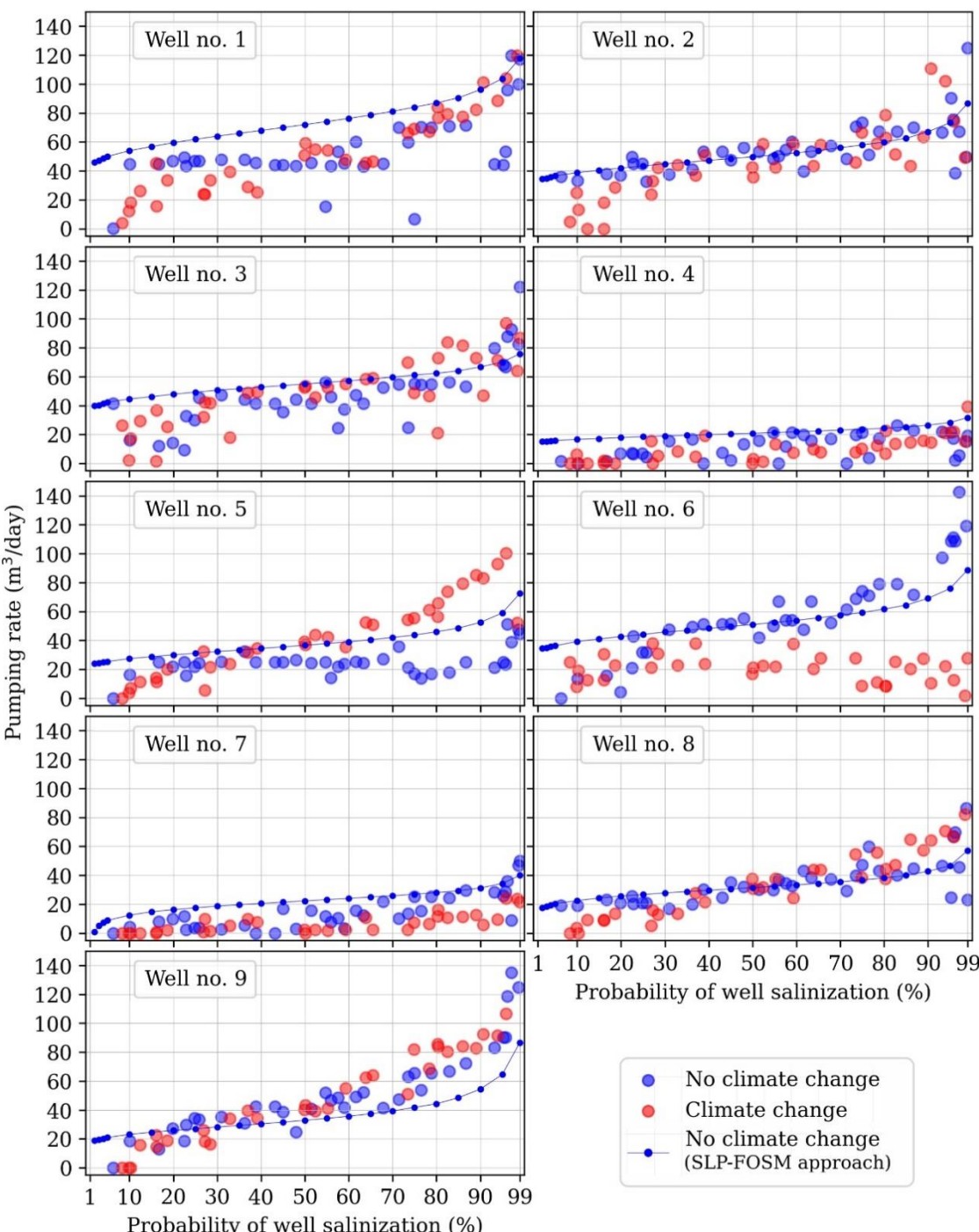

**Figure 11:** Optimal allocation of pumping in the well field as a function of probability of well salinization, when neglecting (blue) or accounting for (red) climate projections (ensemble-based approach). The results obtained using the SLP-FOSM approach (neglecting climate change effects) are also shown. The y-axis was cut off at 150 m³/day.

## 5 Discussion

### 5.1 Optimization under uncertainty

The OUUs identified sets of maximum allowable pumping scenarios corresponding to different probabilities of salinization, from which the groundwater managers can choose depending on their attitude towards risk (Coulon et al., 2022). Different results were obtained when using a MO-ensemble approach or a SLP-FOSM approach to OUU. Since ensemble-based uncertainty

quantification is more reliable than FOSM-based uncertainty estimates, the probabilities of well salinization determined by the MO-ensemble approach are more reliable. In the study area, the SLP-FOSM approach may have overestimated the maximum allowable pumping rates compared to the MO-ensemble approach and underestimated the probability of well salinization associated with pumping scenarios, especially for risk-averse stances. The SLP-FOSM approach would have led to the conclusion that the well field is able to supply both the current water demand and the highest water demand projections, with a wide margin, while maintaining a very low probability of salinization of 2%. In comparison, the MO-ensemble approach found much lower maximum allowable pumping rates, and if the highest risk-averse stance was adopted (i.e., a 6% probability of well salinization), the well field would barely be able to meet the current water demand, and could only meet the higher water demand projections with an increased probability of well salinization. However, in the MO-ensemble approach, the complete convergence to the Pareto front requires many generations of the NSGA-II algorithm, which in practice is limited by computational constraints. Therefore, the maximum allowable pumping rates found by the MO-ensemble approach for the study area may also be suboptimal, and overly conservative.

Incorporating climate projections into the parameter ensembles, instead of moving directly from history matching to OUU, was essential, since neglecting climate projections resulted in underestimating predictive uncertainties (Fig. 9) and therefore underestimating probabilities of well salinization for risk-averse pumping scenarios (Fig. 10). This was the case even though the climate projections hardly affected the median model predictions (Fig. 9): the simple increase in model predictive uncertainty due to additional consideration of climate uncertainty, which can be expected in other areas, was enough to impact the results of the OUU. Accounting for climate projections in the OUU decreased the maximum allowable pumping rates for users with a risk-averse stance, and therefore led to more conservative pumping scenarios. When accounting for climate projections, if the highest risk-averse stance was adopted (i.e., an 8% probability of well salinization), the well field would meet neither the current water demand nor the lowest water demand projections. Therefore, groundwater managers could either decide to meet the water demand but at higher probabilities of salinization (greater than 10–15%), or to find supplementary sources of water. Multiplying the probability of well salinization by its consequences would allow to characterize the risk of salinization, and therefore assist decision-makers in their evaluation of different management options.

While parameter, observation and climate uncertainty were considered in this study, model conceptual uncertainty was neglected; and using a sharp-interface approach to simulate saltwater upconing could result in increased conceptual uncertainty. However, the posterior parameter values were physically plausible and consistent with the prior parameter distributions, and the information in the observations was appropriately assimilated; these are the two indicators available to detect the potential for conceptual model uncertainty issues. The Doherty and Christensen (2011) model pairing methodology could be used to more explicitly investigate the potential for conceptual model issues through pairing of a sharp-interface model with an advective-dispersive-based variable-density model. In the context of lateral seawater intrusion, methodologies have also been developed to optimize pumping using a coupled sharp-interface/advective-dispersive approach (e.g. Christelis and Mantoglou 2018; Dey and Prakash 2022), which could be explored in the context of freshwater lenses. This topic is discussed in more detail in Coulon et al. (2022).

### 5.2 Incorporating climate projections

Although the sea level and recharge projections were considered to be independent (Section 3.3), sea-level rise projections linked to GCMs and RCPs can sometimes be found (e.g., James et al., 2021) and linked sea level-recharge ensembles could be used in future OUUs. However, climate uncertainty is always dependent on the size and nature of the projections contained in the climate ensemble (for example the number and nature of the applied GCMs and RCPs), and the uncertainty is expected to increase with the ensemble size. Even if all available GCMs were used, these would still represent a subset of possible future climates (Ray and

Brown, 2015). Therefore, the optimal pumping scenarios should regularly be updated with the latest climate projections. Another approach could be to integrate robustness to climate uncertainty into the optimization, aiming to find pumping scenarios that are less sensitive to climate uncertainty and that perform well under a wide range of climate projections (Borgomeo et al., 2018).

This study considered climate change effects under steady-state conditions, and explored the management options that would result from the groundwater system (i.e., the freshwater lens) equilibrating with the 2050 climate projections. In reality, since climate change effects are a transient process and this steady state may never be reached, this approach can therefore be viewed as being conservative. When considering transient conditions, climate uncertainties might increase with time and interannual climate variability (e.g. variations in the frequency and intensity of extreme weather events) may also impact the optimization results.

Few examples were found in the existing literature as to how to merge the parameter and climate recharge ensembles (Section 3.3.2). These were obtained independently, through different types of models: a calibrated groundwater flow model with a simplified representation of recharge (MODFLOW-SWI2) and a more complex, uncalibrated groundwater recharge model able to convert weather projections into recharge projections (SWB2). The groundwater recharge model could have been coupled to the groundwater flow model, with its uncertain parameters added to the history matching procedure, and the weather projections would then be run directly through the coupled models. However, the coupling of numerical models is often computationally demanding. In the approach that was selected, the $\Delta R$ ensemble was resampled assuming a normal distribution, as a first approximation, which did not conserve the exact shape of the $\Delta R$ distribution (Fig. 3c, 3d). However, the ensemble-based approach allows for the incorporation of climate projections that do not necessarily follow Gaussian distributions. Rejection sampling or Markov Chain Monte Carlo methods could also be employed to resample the $\Delta R$ ensemble while conserving its shape. Further investigations on this subject would be of interest.

Recharge ($R$) and hydraulic conductivity ($K$) parameters are known to be correlated (Anderson et al., 2015), therefore history matching could have resulted in realizations where high current $R$ is paired with high $K$ values, or low current $R$ is paired with low $K$ values. The current $R$ values stayed paired with their corresponding $K$ field and recharge perturbations $\Delta R$ were randomly added to them. However, no correlation was assumed between current and future recharge. With this assumption, high future $R$ could be paired with low $K$ values, and low future $R$ with high $K$ values; therefore, the tails of the constraint PDFs can be explored more thoroughly. While this assumption might overestimate the constraint uncertainty, it can be viewed as being conservative. Furthermore, bias in future recharge predictions could be caused by having excessive confidence in the $R/K$ correlation learned during history matching.

**5.3 Use of an ensemble-based approach**

Ensemble-based history matching yielded a relatively large range of simulated values for each observation (Fig. 6); however, this could be explained by the cutoff of PESTPP-IES after the second iteration. Over successive PESTPP-IES iterations, the goodness of fit increases and the ensemble diversity (and therefore the posterior parameter variance) decreases (Section 3.2). While it is recommended to use a small number of iterations with the IES algorithm (PEST++ Development Team 2022), the choice of the cutoff iteration can be subjective. For this study, it was decided to maintain a large ensemble diversity (and possibly overestimate posterior parameter variance), rather than taking the risk of underestimating posterior parameter variance and risking biases in the parameter estimates arising from model error. This conservative approach was appropriate since there are no alternative drinking water sources on the island.

Using an ensemble-based approach in combination with a population-based evolutionary algorithm such as NSGA-II required a significant number of model simulations. The use of parallel processing was critical to implement the framework within reasonable computational times and was greatly facilitated by using the PEST++ software, which contains a fault-tolerant, parallel run

manager (White et al., 2020b). However, a compromise had to be made between the number of realizations in the parameter ensemble, the number of individuals in the decision variable population, the frequency at which the prediction ensemble was re-evaluated during the optimization (Section 3.4) and the number of generations of the optimization algorithm. At the end of the optimization, several solutions were dominated by other solutions for both objective functions (Fig. 10 and 11), showing that final convergence to the Pareto front could be further improved. Although complete convergence to the Pareto front was limited by computational constraints, from a practical perspective, the Pareto front that was obtained provides valuable solutions and insights. The Pareto front is sensitive to the size of the parameter ensemble, as shown for example, by Sreekanth et al. (2016) who analyzed convergence to the Pareto front using different numbers of realizations. In particular, the solutions at the extremities of the Pareto front are expected to be sensitive to the tails of the constraint PDFs, but the convergence of the extreme percentiles (e.g. 95th, 99th) of constraint ensembles could require a prohibitive number of realizations. In our case, while the extremities of the Pareto front were found to be sensitive to the ensemble size, using 173 realizations was the maximum that was computationally feasible. Therefore, the highly risk-averse region may not be fully explored. While it is informative, mapping the entire Pareto front is computationally expensive, especially considering that only the high-reliability solutions are generally of interest to groundwater managers. Stack-ordering methods have been developed to find high-reliability solutions in ensemble-based OUUs with a very low computational effort (e.g. Bayer et al., 2010; Paly et al., 2013), the implementation of which would be very useful for managers wishing to adopt highly risk-averse stances. Worst-case scenarios could be explored by running optimizations on the realizations with the lowest current recharge estimates associated with the most extreme recharge decrease scenarios (i.e., the most important $-\Delta R$ perturbations). Finally, with the MO-ensemble approach, the dispersion and non-unicity of the individual pumping rates (i.e., of the decision variables) could make the presentation of the results to decision-makers and their implementation more challenging.

## 6 Conclusion

A fully scripted workflow was developed for ensemble-based history matching, incorporation of climate projections and pumping optimization under uncertainty considering parameter, historical observation and future climate uncertainty. The workflow was implemented in a real-world island aquifer. It allowed for the quantification of the optimal trade-off between pumping and probability of well salinization considering parameter, observation and climate uncertainty simultaneously, letting groundwater managers choose the final pumping scenario depending on their attitude towards risk. Incorporating climate projections into the OUU allows groundwater managers to account for multiple sources of uncertainty simultaneously, i.e., uncertainty arising from the model itself (e.g., parameter and observation uncertainty) and climate uncertainty (e.g., sea level and recharge uncertainty).

The workflow used easily accessible, model-independent tools for ensemble-based history matching and multi-objective optimization under uncertainty, that are applicable to high-dimensional, nonlinear models and to nonlinear optimization problems. The approach can therefore be implemented in a large range of coastal settings, including high-dimensional and nonlinear models, provided that model simulations are parallelized. The workflow could also be adapted to other management problems involving optimization under uncertainty. The multi-objective, ensemble-based approach led to much lower maximum allowable pumping rates than the previously applied sequential linear programming, FOSM-based approach (Coulon et al., 2022), for users with risk averse stances. This was the case even though the numerical model and optimization problem were relatively linear.

A method for merging parameter and climate recharge ensembles was suggested, and the effect of this coupling on management optimization was explored. Incorporating climate uncertainty into the workflow was critical, since it reduced the maximum allowable pumping rate for users with a risk-averse stance, and since neglecting climate uncertainty resulted in underestimating the probabilities of well salinization. In the study area, when considering sea level and recharge projections to the year 2050, the

well field would be unable to meet the current water demand or any of the year 2050 water demand projections while maintaining very low risks of well salinization. Conceptual uncertainty was not considered in the analysis, but its evaluation and coupling with the other sources of uncertainty would be useful as it could also impact the optimization results.

## Code availability

The Python scripts used to implement the workflow and produce figures are available on GitHub at https://github.com/Cecile-A-C/swi2-ensembles and are archived at https://doi.org/10.5281/zenodo.7574457 (version1.0.0, MIT license). MODFLOW-2005 (version 1.12.00) is available at https://www.usgs.gov/software/modflow-2005-usgs-three-dimensional-finite-difference-ground-water-model. PESTPP-IES, PESTPP-SWP and PESTPP-MOU are available at https://www.usgs.gov/software/pest-software-suite-parameter-estimation-uncertainty-analysis-management-optimization-and and the source code is available at

https://github.com/usgs/pestpp/releases (version 5.1.13 was used for PESTPP-IES and PESTPP-SWP and version 5.1.24 was used for PESTPP-MOU).

## Author contribution

**Cécile Coulon**: Conceptualization, Formal analysis, Methodology, Visualization, Writing. **Jeremy T. White**: Formal analysis, Methodology, Software, Writing. **Alexandre Pryet**: Conceptualization, Methodology, Resources, Writing. **Laura Gatel**: Formal

analysis, Methodology. **Jean-Michel Lemieux**: Conceptualization, Resources, Supervision, Writing.

## Competing interests

The authors declare that they have no conflict of interest. Jeremy White was funded by INTERA Incorporated.

## Acknowledgments

The authors would like to thank John Molson for proofreading the manuscript and for his insightful comments, and Sreekanth

Janardhanan and an anonymous reviewer for their valuable comments. This work was funded by the Ministère de l'Environnement et de la Lutte contre les changements climatiques of Quebec, as part of the project « Acquisition de connaissances sur les eaux souterraines dans la région des Îles-de-la-Madeleine » (Groundwater characterization project in the Magdalen Islands region). The authors would like to thank the OURANOS consortium and Marco Braun for supplying the daily precipitation, minimum and maximum air temperature climate simulations for the Magdalen Islands for the period 1950-2100. A scholarship provided by the

Canadian National Chapter of the IAH (IAH-CNC) to Cécile Coulon helped finance this study. Jeremy White was funded by INTERA Incorporated.

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
