# Peer review of "An ensemble-based approach for pumping optimization in an island aquifer considering parameter, observation and climate uncertainty"

_Hydrology and Earth System Sciences, 2023_

## Author Response (AR1)

**Author's response**

The line number(s) in the response are for the TRACK CHANGES version of the revised manuscript.

We thank the reviewers for their valuable comments, which helped us improve the manuscript. Most of the comments aimed at either clarifying the methodology or adding elements to the discussion. All the comments were addressed and the manuscript was reviewed for English.

**Reviewer 1**

In this paper, an approach to optimizing pumping rates on Grande Entrée Island is proposed. This approach considers uncertainty in observations, model parameters, and climate inputs. The manuscript is well-organized, with a clear objective, and the results and conclusions provide answers to the study's objectives. Part of the methodology consists of techniques that were previously proposed and tested in other studies. I think the following comments/suggestions should be addressed:

1. I suggest that the authors review the manuscript for English usage mistakes. While the article is generally well-written, some grammar errors are confusing, and I had to reread parts of the article to fully understand the ideas.

    Changes were made throughout the manuscript. *One main change was modifying "2050 sea-level and recharge ensembles" to "Sea-level and recharge ensembles for the year 2050", to clarify that "2050" relates to the year and not to a number of ensembles. This change was also implemented in the graphical abstract and in Figure 2.*

2. In the first part of the approach, you move from a prior parameter ensemble to a posterior parameter ensemble, using observations. This seems to me like Bayesian model calibration, but you never mention the word "Bayesian" in the manuscript. Why is that? Perhaps you could explain this procedure from a Bayesian perspective.

    History-matching was indeed implemented in a Bayesian framework, using the assumptions of multivariate Gaussian prior and posterior distributions, as encoded within the ensemble-based data assimilation approach. We defined the prior parameter probability distribution from site characterization and expert knowledge and generated an ensemble from the prior distribution using standard sampling techniques. We then conditioned the prior ensemble with the information contained in listed observations by minimizing a model-to-measurement fit objective function (which is inversely proportional to the likelihood function), to yield an ensemble of posterior parameter realizations.

    Changes made (lines 174-180):
    *"**History matching was implemented in a Bayesian framework, using the assumptions of multivariate Gaussian prior and posterior distributions.** Over successive iterations, PESTPP-IES  **conditioned** the prior parameter ensemble  **with the information contained** in 20 freshwater head observations and 142 freshwater-seawater interface elevation observations (derived from deep open wells, TDEM and ERT geophysical surveys, Fig. 1) **by minimizing a model-to-measurement fit objective function. Observations were** paired with random realizations of measurement noise**, and the least-squares objective function was calculated as the sum of squared weighted differences between simulated and observed data** (PEST++ Development Team, 2022)".*

3. Following up on the previous point, is it convenient to distinguish between ensembles and pdfs? Clearly, an ensemble is always different from a PDF. Also, given that the statistical moments of your ensembles sometimes vary by an order of magnitude, what effect do you think this difference will have on the conclusions of your study?

> Thank you for bringing up this interesting point. We agree that the statistical moments of the parameter ensemble and that of the PDF will generally be different, especially for small ensemble sizes. We think it is useful to present both in the manuscript, to show that although ensembles provide more reliable estimations of PDFs than FOSM analysis, they remain samples of, and therefore approximations of the PDFs. The factor that will most influence the outcome of optimization under uncertainty is whether the statistical moments of the constraint ensembles converged relative to the ensemble size, since the probabilities of constraint violation are directly used in the optimization algorithm (Eq 4, 5). We tested prior parameter ensembles with sizes ranging from 50 to 1000 realizations and found that the statistical moments of the 50% seawater salinity ensembles (mean, standard deviation, 5th and 95th percentiles) converged after 200 realizations.

> Changes made (lines 310-312):
> "*As the number of realizations increases, the ensembles become more representative of the PDFs that they sample, but computational times increase. **The most important factor influencing the outcome of the OUU procedure is whether the statistical moments of the constraint ensembles have converged relative to the ensemble size, since the probabilities of constraint violation are directly used in the optimization algorithm (Eq 4, 5).** Analyzing the convergence of posterior $\zeta_{50\%}$ ensemble mean and standard deviation values as a function of prior ensemble size led to the selection of a prior parameter ensemble containing 200 realizations.*"

4. How do you perform the sampling? Do you assume that your parameters are mutually independent?

> See # 6.

5. Why were the initial transition zone width and longitudinal dispersivity not considered in the first part of the approach? I see that you calibrated 58 parameters, so what is the reason for not including the other two parameters?

> In the same study area, the previous deterministic parameter estimation in Coulon et al (2021) did not consider the initial transition zone width $M$ and longitudinal dispersivity $\alpha_L$, and these parameters were only considered for the FOSM-based optimization under uncertainty (Coulon et al 2022). In order to compare the ensemble-based approach to the deterministic approach, we decided to conserve this strategy. Furthermore, $\alpha_L$ and $M$ are uninformed by the head and interface observations used in history matching (but may influence the predicted simulated salinity contours).

> Changes made (lines 168-170):

> "*Several constant model parameters were included in this ensemble and remained fixed during history matching, including the sea level, longitudinal dispersivity $\alpha_L$ and the initial width of the transition zone $M$ (Table 2). **The dispersivity $\alpha_L$ and transition zone width M were fixed during history matching but adjustable during optimization, to be consistent with the previous SLP-FOSM approach and to enable a comparison between both approaches.**"*

6. In the sentence that starts on line 152, you write that you account for spatial correlations between pilot points. However, I think this part is not well-explained in the document. Are you using kriging somehow? Or does it mean that the samples you draw from a parameter x_i are conditioned on the values of a parameter x_j? Please explain the method you use to account for spatial correlation.

> Answers to 4 and 6: Random parameter fields were sampled from the prior parameter PDFs, assuming they can be described by multi-Gaussian distributions and using a prior parameter covariance matrix where the diagonal elements contain the variances of the parameters, and the off-diagonal elements contain the covariances. Note that parameters varying over several orders of magnitude were all log-transformed. It was further assumed that all parameters were statistically independent (all off-diagonal elements are null), with the exception of pilot point parameters which were spatially correlated (non-null off-diagonal elements). An exponential variogram with a range equal to 3 times the pilot point spacing (i.e., 500 m) was used to describe the spatial correlation between the hydraulic conductivities at pilot point locations. We also note that as part of the pilot-point parameterization, ordinary kriging was used to interpolate grid cell values from pilot point locations to the model cells.

> Changes made: sentence added (lines 125-126):
> *"As part of the pilot-point parameterization, ordinary kriging was used to interpolate grid cell values from pilot point locations to the model cells."*

> And lines 159-166:
> *"N$_{prior}$ **random** realizations were drawn from the prior probability distribution functions (PDFs) of these parameters **(Table 1), assuming they could be described by multi-Gaussian distributions and using a prior parameter covariance matrix. It was assumed that all parameters were statistically independent, except for pilot point parameters which were spatially correlated. To describe the spatial correlation between the hydraulic conductivities at pilot point locations, an exponential variogram with a range equal to 3 times the pilot point spacing (i.e., 500 m) was used. We note that parameters varying over several orders of magnitude were all log-transformed.** "*

7. On line 166, you mention that 500-year simulations were carried out. Why 500? Why not 100 or 1000?

> We agree that this is not clear in the manuscript. We ran the simulations until heads and interface elevations were stable close to pumping wells, and initial testing showed this was achieved within 500 years.

> Changes made (lines 183-185):
> *"History matching was conducted under steady-state conditions, using **long transient (**500-year**)** simulations with constant boundary conditions (i.e., sea level, recharge and pumping rates) representative of the average conditions during the 2014–2019 calibration period. **The simulations were run until heads and interface elevations close to pumping wells were stable, which was achieved within 500 years.**"*

8. Could you please include the mean absolute error metric in the results of Figures 6a and 6b? In Fig. 6b, the vertical axis refers to the residual (Y_sim – Y_obs). Isn't this residual large? In the caption, it is written that it is a scatter plot, so which is correct?

The y axis of Figure 6b should also be labeled "Simulated (m)", thank you for finding this error. The range of simulated values is large for each observation; however, this is partly explained by the cutoff of PESTPP-IES after iteration 2. Over successive IES iterations, the goodness of fit increases and the ensemble diversity (and therefore the posterior parameter variance) decreases. Using a small number of iterations is recommended when using IES, but we acknowledge that the choice of the cutoff iteration is subjective. We preferred to maintain a large ensemble diversity and possibly overestimate posterior parameter variance, rather than taking the risk of underestimating posterior parameter variance and/or risking biases in the parameter estimates arising from model error phenomena. We thought being conservative was appropriate since there are no alternative drinking water sources on the island and the consequences of well salinization can be long-lasting.

Changes made in Figure 6: *MAE metrics are added and the vertical axis of Figure 6b is renamed* ***"Simulated (m)"***

Sentence added to the caption (lines 329-330): "***The average MAE value (mean average error) for each observation group is shown.***"

And discussion added (lines 470-477):
*"**Ensemble-based history matching yielded a relatively large range of simulated values for each observation (Fig. 6); however, this could be explained by the cutoff of PESTPP-IES after the second iteration. Over successive PESTPP-IES iterations, the goodness of fit increases and the ensemble diversity (and therefore the posterior parameter variance) decreases (Section 3.2). While it is recommended to use a small number of iterations with the IES algorithm (PEST++ Development Team 2022), the choice of the cutoff iteration can be subjective. For this study, it was decided to maintain a large ensemble diversity (and possibly overestimate posterior parameter variance), rather than taking the risk of underestimating posterior parameter variance and risking biases in the parameter estimates arising from model error. This conservative approach was appropriate since there are no alternative drinking water sources on the island.**"*

9. What does the top horizontal axis, "Probability density," mean in Figure 10?

   This wasn't clear, the top horizontal axis is the probability density function associated with the 2050 water demand.

   Changes made in Figure 10: *the top horizontal axis of Figure 10 is renamed "**Probability density function of the projected water demand**"*

   And in its caption (lines 390-391): "*The current water demand and the **probability density function associated with the**  projected **(year 2050)** water demand are superimposed.*"

10. In your study, you consider three sources of uncertainty (observations, model parameters, and climate forcing). However, you always rely on the same numerical model, which is not very accurate (according to Figure 6). Is the model's conceptualization another source of uncertainty? How would you consider this source of uncertainty? I don't expect you to include this analysis in the current

study, which is already very comprehensive, but I wanted to highlight this source of uncertainty for your future studies.

> We agree that the model conceptualization is an additional source of uncertainty. This is discussed in more detail in Coulon et al (2022) but should be mentioned in this paper as well. Using a sharp interface approach was a simplification of mixing processes which could result in increased conceptual uncertainty. However, the posterior parameter values display physically plausible values that are coherent with prior parameter distribution and the information in the observations was assimilated in appropriate ways; these are the two indicators available to detect the potential for conceptual model uncertainty issues. The Doherty and Christensen (2011) (https://doi.org/10.1029/2011WR010763) model pairing methodology could be used to more explicitly investigate the potential for conceptual model issues surrounding the use of the sharp-interface approximation through pairing with an advective-dispersive-based variable-density model. In the context of lateral seawater intrusion, methodologies have also been developed to optimize pumping using a coupled sharp-interface/advective-dispersive approach (e.g. Christelis and Mantoglou 2018 - https://doi.org/10.1007/s11269-018-2116-0; Dey and Prakash 2022, https://doi.org/10.1007/s11269-022-03145-w) and this could be explored in the context of freshwater lenses.
>
> Changes made: sentenced removed (lines 399-400):
> *"The limitations associated with using a sharp-interface model to simulate saltwater upconing are discussed in detail in Coulon et al. (2022)."*
>
> And discussion added (lines 426-434):
> ***"While parameter, observation and climate uncertainty were considered in this study, model conceptual uncertainty was neglected; and using a sharp-interface approach to simulate saltwater upconing could result in increased conceptual uncertainty. However, the posterior parameter values were physically plausible and consistent with the prior parameter distributions, and the information in the observations was appropriately assimilated; these are the two indicators available to detect the potential for conceptual model uncertainty issues. The Doherty and Christensen (2011) model pairing methodology could be used to more explicitly investigate the potential for conceptual model issues through pairing of a sharp-interface model with an advective-dispersive-based variable-density model. In the context of lateral seawater intrusion, methodologies have also been developed to optimize pumping using a coupled sharp-interface/advective-dispersive approach (e.g. Christelis and Mantoglou 2018; Dey and Prakash 2022), which could be explored in the context of freshwater lenses. This topic is discussed in more detail in Coulon et al. (2022)."***

In addition, please consider the following minor comments.

1. Define the acronyms TDEM and ERT in figure 1.

   Changes made (line 109):
   ***"(including electrical resistivity tomography and time-domain electromagnetic surveys)"***

2. Caption of figure 1 says "The BC implemented in MODFLOW are shown" but they are not shown in the figure, they are described in the caption. Please rephrase.

   Changes made (line 110-112):

*"The boundary conditions implemented in MODFLOW are  a uniform recharge rate on land cells (RCH package), general head boundary conditions for sea cells (GHB package), groundwater pumping at municipal wells (MNW2 package)."*

3.  In line 334 you use the words "former" and "latter". However, given that in the same section you speak about the results of your current study and the results of your previous study, the usage of those words can be confusing. Could you consider using other words?

Changes made (lines 367-369):
*"For example, a 230 $m^3$/day pumping rate corresponded to a 25% probability of salinization using the  **MO-ensemble** approach, versus 2% using the  **SLP-FOSM** approach."*

**Reviewer 2**

Overview:

Authors of the manuscript "An ensemble-based approach for pumping optimization in an island aquifer considering parameter, observation and climate uncertainty" applied a multi-objective optimisation under uncertainty approach available through PEST-MOU tool kit to find Pareto-optimal solutions for management of sea water intrusion in a coastal island considering maximisation of pumping and reliability of meeting the salinity constraint as conflicting objectives. The manuscript is well-written and well-organised. It is a contribution of interest to readers of HESS and advances the science in methodologies for optimising groundwater management. I have only a few minor comments and hence recommend accepting the manuscript after minor revisions.

We thank Dr Sreekanth Janardhanan for his comments, which we have addressed in the final draft. Here is our response to the comments:

Individual comments:

*   Section 3.3: It is not clear if the climate change predictive runs were also steady-state? Could you please explain that in this section.

    Climate change predictive simulations were conducted under steady-state conditions, with boundary conditions representative of year 2050 sea-level and recharge conditions. All predictive simulations were conducted under steady-state conditions because the storage parameters were unconstrained by the history matching, and therefore remained highly uncertain. This was added to the methods. As mentioned in the discussion, since climate change effects are a transient process and this steady state may never be reached, this approach can be viewed as being conservative.

    Changes made (lines 201-204):
    *"**The climate change predictive simulations were then conducted under steady-state conditions, with boundary conditions representative of 2050 sea-level and recharge conditions. All predictive simulations were conducted under steady-state conditions because the storage parameters were unconstrained by the history matching, and therefore remained highly uncertain.**"*

*   Line 210: It is not very clear how values are resampled. Could you please explain.

- Line 215: Ah OK. So the  are uncorrelated with current recharge – does this mean  were randomly added to realisations in current recharge ensemble ?

   The recharge perturbations $\Delta R$ were indeed uncorrelated with the current recharge values $R_{current, MODFLOW}$. The $\Delta R$ and $R_{current, MODFLOW}$ realizations were randomly paired together to generate the $R_{2050, MODFLOW\ ensemble}$. This is reformulated in the manuscript.

   Changes made (lines 234-236):
   "**The $\Delta R$ and $R_{current, MODFLOW}$ realizations were then randomly paired together to generate the $R_{2050, MODFLOW\ ensemble}$**  ensembles"

- Line 245: 500 year initial simulations with zero pumping" – Is this for reaching a new equilibrium corresponding to the new recharge and other factors corresponding to future climate? I assume this steady state can be considerably different from the historical steady state with recharge and sea level changes?

   Initial, steady-state simulations with no pumping were run until heads and interface elevations were stable close to pumping wells. Initial testing showed this was achieved within 500 years. As shown in Figure 9, there is variability both in the steady states obtained with the posterior ensemble and the 2050 predictive ensemble. For the interface elevation under pumping wells, the mean steady-state value is similar in both cases, but there is twice as much variability in the initial steady states when climate change is accounted for.

   Changes made (lines 273-275):
   "**Long transient (**500-year**)** initial simulations with  **no** pumping were run through both parameter ensembles,  **to allow the freshwater lens to reach a steady state under the climate forcings and hydraulic properties prescribed in** each realization."

   And sentence added (lines 342-343):
   "**Figure 9 shows the variability in the pumping optimization initial conditions obtained with both ensembles.**"

- Line 250: What is the significance of 200-year simulation period?

   The pumping optimization under uncertainty was conducted under steady-state conditions, using long transient simulations to allow the freshwater lens to reach a new steady state under the pumping rates tested. This new steady state was achieved within 200 years.

   Changes made (lines 275-277):
   "The pumping optimization under uncertainty was then conducted under steady-state conditions**, using long transient (200-year) simulations to allow the freshwater lens to reach a new steady state under the tested pumping rates**  **The** occurrence of well salinization was examined at the end of the 200-year simulation period."

- Line 265: "Prediction ensemble was reevaluated every 10 generations and reused in intermediate generations" – how was objective function calculated in the intermediate generations?

In the version 5.1.24 of PESTPP-MOU that was used, each individual in intermediate generations was mapped to the nearest individual, in decision-variable space, at which constraint probability distribution functions (PDFs) were previously evaluated, in a minimum-Euclidean-distance sense. The constraint PDFs of the latter were translated to the former, by differencing the simulated constraint values between the two individuals, assuming these values represented the mean of the PDFs. This approach assumes that individuals close to each other in decision variable space have similar constraint PDFs (PEST++ Development Team, 2022; White et al., 2022).

Changes made (sentence added lines 294-297):
*"In intermediate generations, each individual was mapped to the nearest individual at which constraint PDFs had been previously evaluated, in a minimum-Euclidean-distance sense, and the constraint PDFs of the latter were translated to the former. This approach assumes that individuals close to each other in decision variable space have similar constraint PDFs (PEST++ Development Team, 2022; White et al., 2022)."*

- Figure 7: Figure 7 shows considerable correlation between recharge and K values. Perhaps discuss the implication of this in predictive simulations with a different recharge regime (although noting that the mean doesn't change much) that does not consider correlations with historical recharge. Is it likely to bias predictions?

Parameter correlations can be identified during history matching. Recharge ($R$) and hydraulic conductivity ($K$) parameters are known to be correlated (Anderson et al., 2015), which can result in realizations where high current $R$ is paired with high $K$ values or low current $R$ is paired with low $K$ values. The current $R$ values stayed paired with their corresponding $K$ field and recharge perturbations due to climate change ($\Delta R$) were randomly added to them. However, no correlation was assumed between current and future recharge. With this assumption, high future $R$ (i.e., high $\Delta R$) could be paired with low $K$ values, and low future $R$ (i.e., low $\Delta R$) with high $K$ values; therefore, the tails of the constraint PDFs can be explored more thoroughly. While this assumption might overestimate the constraint uncertainty, it can be viewed as being conservative. Furthermore, bias in future recharge predictions could be caused by having excessive confidence in the $R/K$ correlation learned during history matching.

Changes made (lines 461-468):
*"Recharge (R) and hydraulic conductivity (K) parameters are known to be correlated (Anderson et al., 2015), therefore history matching could have resulted in realizations where high current R is paired with high K values, or low current R is paired with low K values. The current R values stayed paired with their corresponding K field and recharge perturbations ΔR were randomly added to them. However, no correlation was assumed between current and future recharge. With this assumption, high future R could be paired with low K values, and low future R with high K values; therefore, the tails of the constraint PDFs can be explored more thoroughly. While this assumption might overestimate the constraint uncertainty, it can be viewed as being conservative. Furthermore, bias in future recharge predictions could be caused by having excessive confidence in the R/K correlation learned during history matching."*

- Figures 10 and 11: Looks like the final convergence of the Pareto-optimal front hasn't been achieved. There are several solutions that are dominated by other solutions for both objective functions. I assume you had to optimise the number of iterations (150) and population (30) of NSGA-II to make it computationally feasible and it may have affected the pareto-optimality? From a practical point of view, these still give valuable solutions.
- Line 375: Ah, I see the answer to the above question explained here.

Changes made (lines 481-486):
*"However, a compromise had to be made between the number of realizations in the parameter ensemble, the number of individuals in the decision variable population, the frequency at which the prediction ensemble was re-evaluated during the optimization (Section 3.4) and the number of generations of the optimization algorithm. At the end of the optimization, several solutions were dominated by other solutions for both objective functions (Fig. 10 and 11), showing that final convergence to the Pareto front could be further improved. Although complete convergence to the Pareto front was limited by computational constraints, from a practical perspective, the Pareto front that was obtained provides valuable solutions and insights."*

- line 420 – 425: Yes, stack ordering approaches may relieve the computational burden. Especially in this case, if you pick realisations from the tail end of recharges for worst climate scenarios it would do the job?

We agree that worst-case scenarios could be explored by running optimizations on the realizations with the lowest historical recharge values associated with the most extreme recharge decrease scenarios (i.e., the most important -$\Delta R$ perturbations). However, realizations remain samples of PDFs, and the $\Delta R$ realizations randomly sampled from the $\Delta R$ probability distribution function may not be representative of its extreme percentiles. Therefore, these worst-case realizations could still underestimate the worst-case scenario.

Changes made (lines 497-498):
*"Worst-case scenarios could be explored by running optimizations on the realizations with the lowest current recharge estimates associated with the most extreme recharge decrease scenarios (i.e., the most important -$\Delta R$ perturbations)."*

**Other changes**

- Affiliation: the affiliation (1) was also added to co-author Alexandre Pryet's name.

- Acknowledgments added (lines 539-540):
*"The authors would like to thank John Molson for proofreading the manuscript and for his insightful comments, and Sreekanth Janardhanan and an anonymous reviewer for their valuable comments."*

- References: the reference list was updated with the additional references